EMBO
Molecular Medicine

# *Salmonella* cancer therapy metabolically disrupts tumours at the collateral cost of T cell immunity

Alastair Copland [1]✉, Gillian M Mackie [1], Lisa Scarfe[1], Elizabeth Jinks[1], David A J Lecky[1], Nancy Gudgeon[1,2], Riahne McQuade[1], Masahiro Ono [3], Manja Barthel [4], Wolf-Dietrich Hardt[4], Hiroshi Ohno [5,6], Wilma H M Hoevenaar [7], Sarah Dimeloe[1,2], David Bending [1] & Kendle M Maslowski [1,2,8,9]✉

## Abstract

Bacterial cancer therapy (BCT) is a promising therapeutic for solid tumours. *Salmonella enterica* Typhimurium (STm) is well-studied amongst bacterial vectors due to advantages in genetic modification and metabolic adaptation. A longstanding paradox is the redundancy of T cells for treatment efficacy; instead, STm BCT depends on innate phagocytes for tumour control. Here, we used distal T cell receptor (TCR) and IFNγ reporter mice (*Nr4a3*-Tocky-*Ifnγ*-YFP) and a colorectal cancer (CRC) model to interrogate T cell activity during BCT with attenuated STm. We found that colonic tumour infiltrating lymphocytes (TILs) exhibited a variety of activation defects, including IFN-γ production decoupled from TCR signalling, decreased polyfunctionality and reduced central memory ($T_{CM}$) formation. Modelling of T-cell–tumour interactions with a tumour organoid platform revealed an intact TCR signalosome, but paralysed metabolic reprogramming due to inhibition of the master metabolic controller, c-Myc. Restoration of c-Myc by deletion of the bacterial asparaginase *ansB* reinvigorated T cell activation, but at the cost of decreased metabolic control of the tumour by STm. This work shows for the first time that T cells are metabolically defective during BCT, but also that this same phenomenon is inexorably tied to intrinsic tumour suppression by the bacterial vector.

Keywords *Salmonella*; Cancer Therapy; T Cells; Immunometabolism; Asparagine
Subject Categories Cancer; Immunology

## Introduction

Bacterial cancer therapy (BCT) is the ancestor of modern immunotherapy, having origins in the late 19th Century, during which time U.S. physician William Coley used injections of *Streptococcus* to induce tumour regression in inoperable patients (Starnes, 1992). Yet it is only recently that there has been such a sharp resurgence of interest in BCT, owing to significant advances in genetic modification of bacterial vectors, coupled with safety and efficacy concerns regarding immune checkpoint blockade (ICB).

*Salmonella enterica* serovar Typhimurium (STm) is among the most-studied bacterial vectors for cancer therapy. As a metabolic 'generalist' (Dandekar et al, 2014), the bacterium can rapidly adapt to harsh tumour microenvironments, alongside potent tumour-homing abilities. We recently demonstrated in colorectal cancer (CRC) models that an attenuated STm mutant can reduce colonic tumour burden in vivo by imposing strong metabolic competition on the tumour, with a particular reduction in amino acids, sugars and tricarboxylic acid (TCA) cycle intermediates, as well as specific targeting of tumour stem cells resulting in the normalisation of tissue ontogeny (Mackie et al, 2021). The immunological consequences of these changes, however, are hitherto unexplored.

It is widely reported that STm induces a cascade of immune changes within the tumour microenvironment—such as widespread immune cell influx, pro-inflammatory cytokine production, dendritic cell (DC) maturation and natural killer (NK) cell activation—that is believed to support immune-mediated tumour clearance (Yang et al, 2021). Indeed, monocyte depletion (Johnson et al, 2021), NK cell depletion (Lin et al, 2021), or *MyD88* deletion (Zheng et al, 2017) can wholly abrogate therapeutic success in vivo, underscoring the necessity of an innate-like response in this therapy.

Paradoxically, T cells appear to play little role in *Salmonella* BCT, despite their established role in tumour control. It has been reported for over two decades that tumour control is proportionally identical between wild-type (WT) mice and athymic nude/SCID

[1]Institute of Immunology and Immunotherapy, University of Birmingham, Birmingham B15 2TT, UK. [2]Institute for Metabolism and Systems Research, University of Birmingham, Birmingham B15 2TT, UK. [3]Department of Life Sciences, Imperial College London, London SW7 2AZ, UK. [4]Institute of Microbiology, Department of Biology, ETH Zürich, Zürich 8093, Switzerland. [5]Laboratory for Intestinal Ecosystem, RIKEN Institute for Integrative Medical Science, Yokohama, Japan. [6]Immunobiology Laboratory, Graduate School of Medical Life Science, Yokohama City University, Yokohama, Japan. [7]Cancer Research UK Scotland Institute, Garscube Estate, Switchback Road, Glasgow G61 1BD, UK. [8]Present address: Cancer Research UK Scotland Institute, Garscube Estate, Switchback Road, Glasgow G61 1BD, UK. [9]Present address: School of Cancer Sciences, University of Glasgow, Garscube Estate, Switchback Road, Glasgow G61 1QH, UK. ✉E-mail: a.copland@bham.ac.uk; kendle.maslowski@glasgow.ac.uk

mice after *Salmonella* treatment (Kaimala et al, 2014; Lin et al, 2021; Luo et al, 2001); indeed, many efficacy studies are conducted in athymic mice with xenograft tumours (Zhao et al, 2005; Zheng et al, 2017). Antibody depletion of CD4/CD8 T cells does not fully prevent *Salmonella*-mediated reduction in tumour volume (Lee et al, 2011), and adoptive transfer of anti-tumour, STm-generated T cells show no therapeutic capacity (Stark et al, 2009). Most strikingly, mice which have been cured of tumours by STm treatment do not possess an anti-tumour memory response upon re-challenge (Yoon et al, 2017). Taken together, T cells are likely redundant for *Salmonella* BCT.

In this study, we combined an autochthonous colitis-associated CRC mouse model using STm$^{\Delta aroA}$ (an attenuated aromatic amino acid auxotrophic mutant) with the *Nr4a3*-Tocky-*Ifng*-YFP kinetic TCR-cytokine reporter, which has recently been utilised to detect favourable responses to ICB (Elliot et al, 2021). This allowed unparalleled resolution of T cell activation dynamics during STm BCT. We found that T cells were indeed dysfunctional during STm BCT, showing profound decoupling of several activation nodes. This was not associated with any detectable defects in proximal T cell receptor (TCR) signalling, but instead a potent metabolic disruption and inability to sustain a conventional activation trajectory. T cell metabolic dysfunction was due solely to asparagine depletion by bacteria, leading to depleted c-Myc protein; reversal of this depletion restored T cell function. Critically, STm-mediated c-Myc suppression was also detected in the tumour itself, which dampened tumour stemness and survival, highlighting an important 'double-edged sword' for STm BCT in which tumour control by bacteria comes at the detriment of adaptive immunity. In vivo, STm lacking the asparaginase still controls tumour growth. These findings provide a strong rationale for addressing a previously unknown cardinal defect in *Salmonella*-based cancer therapies to yield more successful clinical outcomes.

## Results

### *Salmonella* cancer therapy induces T cells with an aberrant activation signature

T cells are reported as redundant for *Salmonella*-based cancer therapies (Kaimala et al, 2014; Lin et al, 2021; Luo et al, 2001), and so we questioned whether they possessed canonical activation during BCT. To address this, we utilised an autochthonous colitis-associated colorectal cancer model (CAC) in *Nr4a3*-Timer-of-cell-kinetics-and-activity-*Ifng*-YFP ('Tocky-GreatSmart') (Fig. 1A) mice. These mice contain a downstream *Nr4a3* TCR reporter (Bending et al, 2018a, 2018b), which expresses a fluorescent Timer protein (Subach et al, 2009) in response to TCR signalling. After being expressed, the chromophore of the Timer protein matures from blue to red form ($t_{1/2} = 4$ h), reflecting the downstream activation of *Nr4a3* in response to TCR signalling and providing temporal insights into T cell activation. This was used in conjunction with an *Ifng*-YFP reporter (Price et al, 2012), in order to probe T cell activity in tumours and lymphatic tissues (gating strategy shown in Appendix Fig. S1). Following tumour induction, mice were orally gavaged with two doses (once per week) of an auxotrophic STm mutant (STm$^{\Delta aroA}$), which we and others have previously reported to reduce tumour burden in cancer models (Fensterle et al, 2008; Johnson et al, 2021; Mackie et al, 2021).

Flow cytometric analysis of mesenteric lymph nodes (mLN), spleen and tumour revealed that, as previously reported in other cancer models (Lee et al, 2008; Lin et al, 2021), *Salmonella* treatment induced high production of IFN-γ across all tissues in CD4 and CD8 T cells compared to the PBS control (Fig. 1B). Strikingly, however, parsing of TCR reporter expression in reactive T cells showed that there was a strong bias towards a decoupling of IFN-γ expression from the evidence of recent TCR signals, with STm generating significantly more IFN-γ$^{+}$*Nr4a3*-Timer$^{neg}$ T cells than the PBS control group (Fig. 1C). Given that cytokine production and TCR signalling were being dissociated, we next tested whether the extent of IFN-γ expression (measured by MFI) correlated with the likelihood of TCR engagement (measured by % *Nr4a3*-Timer$^{+}$) in activated IFN-γ$^{+}$ TILs participating in the response. As shown in Fig. 1D, T cells in mice treated with PBS had a much stronger correlation between TCR engagement and magnitude of IFN-γ production, whereas *Salmonella* induced a pronounced decoupling of these aspects of T cell activation ($p = 0.0002$ PBS vs STm, pooled T cells).

To assess whether TCR engagement was indeed hampered during bacterial infection, PD-1 was used as a marker of TCR triggering. PD-1 is under strict control of the transcription factor NFAT (Martinez et al, 2015), which translocates to the nucleus upon TCR ligation. During T cell activation, PD-1 is usually co-expressed with *Nr4a3* (Elliot et al, 2021). As expected, PD-1 expression in both groups correlated tightly with evidence of TCR engagement (Appendix Fig. S2). Crucially, however, IFN-γ$^{+}$ TILs in *Salmonella*-treated mice exhibited a reduction in expression of PD-1 when compared to the PBS control group, in keeping with the TCR reporter showing less evidence of TCR engagement (Fig. 1E), as sustained TCR engagement is necessary for upregulation of PD-1 (Elliot et al, 2021).

### T cells from infected tumours are hyporesponsive to conventional TCR-based activation

Given that T cells in STm-based BCT were being unconventionally activated, we hypothesised that T cells would be refractory to TCR stimulation. To this end, we employed a recently described ex vivo tumour platform (Voabil et al, 2021). Primary colonic tumour fragments were isolated and cultured in Matrigel to preserve the tumour architecture and microenvironment, thus permitting synchronous and polyclonal interrogation of activation dynamics. Tumours from PBS and *Salmonella*-treated mice were subjected to α-CD3 and α-CD28 stimulation ex vivo, and *Nr4a3*-Timer was measured in T cells, with the hypothesis of dysfunctional activation (Fig. 2A). CD4 T cells were analysed as they were the most abundant T cell type; CD8 T cells were scarce after culture in Matrigel (Appendix Fig. S3).

As can be seen in Fig. 2B, polyclonal activation of CD4 T cells in primary tumours led to blunted *de novo Nr4a3*-Timer Blue expression, which is expressed immediately after triggering the TCR complex. CD4 TILs from PBS-treated animals were able to upregulate *Nr4a3*-Timer Blue (~40% increase, $p < 0.01$), whereas there was no significant change for the STm group. Next, we wished to know whether the function was also affected. T cell cytokine polyfunctionality is a hallmark of optimal TCR engagement, and is beneficial against tumours and infectious diseases (Hart et al, 2018; Zhao et al, 2016). Tumour fragments were, therefore, activated and

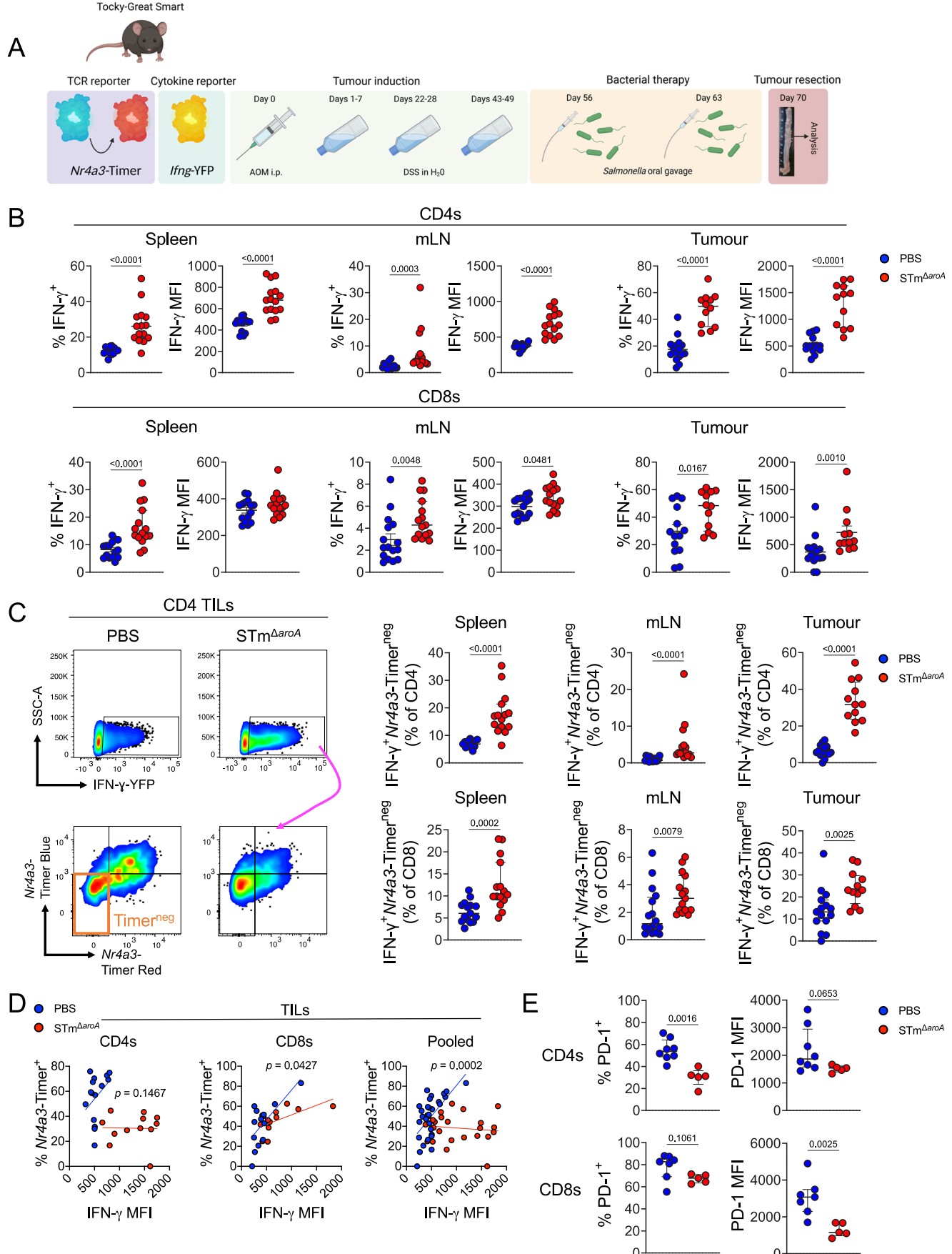

**Figure 1.  Attenuated *Salmonella* cancer therapy induces aberrant T cell activation that decouples IFN-γ from TCR signalling.**

(A) Schematic depicting the induction of colonic tumours in mice by the CAC model using azoxymethane (AOM) and dextran sodium sulfate (DSS) and treatment with attenuated *Salmonella*. (B) Flow cytometric analysis of IFN-γ-YFP in CD4 and CD8 T cells, across spleen, mLN and tumour. Percentage positive and median fluorescence intensity (MFI) of the positive fraction are depicted. (C) Left: representative flow cytometry plot showing gating on IFN-γ[+] T cells and sub-analysis of *Nr4a3*-Timer expression (Blue or Red form). Right: analysis of the IFN-γ[+] and Timer[neg] cells as a fraction of either CD4 or CD8 T cells. (D) Analysis of tumour IFN-γ[+] T cells and a correlation between IFN-γ intensity (MFI) and frequency of TCR signalling (%*Nr4a3*-Timer[+]). The line depicts simple linear regression. (E) Flow cytometric analysis of PD-1 expression (percentage and MFI) within tumour IFN-γ[+] T cells. Bars median average with interquartile range. Each point represents one mouse. (B–D) Spleen: PBS $n = 15$, STM $n = 15$; mLN: PBS $= 16$, STm $= 15$; Tumour CD4s: PBS $= 16$, STm $= 12$; Tumour CD8s: PBS $= 14$, STm $= 12$. E PBS $n = 8$, STm $n = 5$. Significance was tested by unpaired Mann–Whitney (B, C, E) or by linear regression of the two curves (D). Data were pooled from two experiments (B–D).

tested for concurrent Th1 cytokine production. As shown in Fig. 2C, CD4 T cells from the STm-infected mice showed an overall blunted production of TCR-driven cytokine production, with fewer triple- and double-cytokine positive cells when analysed qualitatively (Fig. 2D). These data confirmed that T cell activation was being actively blunted by *Salmonella* in the context of cancer therapy.

## T cell dysfunction is replicated by *Salmonella*-infected tumour organoids

We have previously observed strong concordance between the changes caused by STm in vivo and in vitro, using tumour organoids to explore changes in the metabolome and stem cell compartment during BCT (Mackie et al, 2021). Tumour organoids (small intestinal and colonic) were therefore generated and infected with STm for up to 48 h, including a gentamicin wash-supplement step to kill bacteria outside of the intracellular tumour niche. This allowed for splenocytes and T cells to be directly co-cultured and assessed for activation within the tumour microenvironment (TME), or co-cultured with tumour-conditioned media (TCM) (Fig. 3A). Since microbial TLR ligands can positively or negatively affect lymphocyte activation (Srinivasan and McSorley, 2007; Sturm et al, 2005), heat-killed (HK) STm was used as a control at an equivalent concentration to the live bacteria. T cell activation was assessed by the flow cytometric location and intensity of the *Nr4a3*-Timer (Fig. 3B, Bending et al, 2018a).

As expected, T cells activated in the presence of non-infected or heat-killed STm-treated tumours were able to fully activate, as indicated by the high expression of both Timer Blue and Timer Red (>80% Timer Blue[pos]Timer Red[pos], i.e. persistent TCR signalling), and only minor levels of Timer Blue[neg]Timer Red[pos] (arrested TCR signalling) cells (Fig. 3C). Strikingly, T cells activated in the presence of tumours infected with live *Salmonella* showed a pronounced failure to sustain activation (>40%, $p < 0.0001$ live STm vs non-treated and HK STm at 24 h) (Fig. 3C). This aborting of T cell activation was similarly reflected in the lower fluorescence intensity of Timer Blue and Timer Red, as measured by MFI (Fig. 3D). The benefit of the Timer system is that you can gain temporal insight into activation dynamics. As a comparison we used the widely available Nr4a1-GFP mouse (Moran et al, 2011), and the Nr4a1-Tempo (Elliot et al, 2022) mouse to control for the change in Nr4a-family reporter gene. Nr4a1-Tempo splenocytes showed the same termination of activation when cultured with STm-treated TCM as for the Nr4a3-Timer splenocytes (Fig. EV1A). In contrast, the Nr4a1-GFP system shows equivalent induction of GFP with NT TCM and with STm TCM (Fig. EV1B). This direct

comparison demonstrates the utility of a time-sensitive system; with only GFP reporting, you would conclude that cells are activating in an equivalent manner. This has been used to show the termination of signalling following withdrawal of stimulus, and re-activation following re-stimulation (Elliot et al, 2021; Elliot et al, 2022), but this is the first report to show the Timer system capturing aborted activation.

TCR signalling is intimately connected with cytokine production (Tao et al, 1997), and so we next assessed IFN-γ expression. As with TCR signalling, T cells activated in the milieu of uninfected or HK STm-treated tumours were able to produce high levels of IFN-γ (~50% IFN-γ[+] CD8s at 48 h), but there was a dampening of cytokine expression in lymphocytes exposed to STm-harbouring tumours ($p < 0.0001$ live STm vs NT and HK STm at 48 h), particularly in CD8 T cells (Fig. 3E). This was not limited to type 1 cytokines, as IL-2 secretion was similarly affected in T cells cultured with infected tumours (Fig. 3F). Since IL-2 is a key cytokine for T cell proliferation and growth, we next analysed cell size and division after 3 days of culture. As shown in Fig. 3G, CD8 T cells in the live STm group were smaller after stimulation, suggesting a lower proportion of cells undergoing mitosis. In accordance, these cells were unable to divide, as measured by dilution of tracer dye ($p > 0.05$, live STm unstimulated vs stimulated) (Fig. 3G). Taken together, STm-infected tumours can disrupt multiple facets of T cell activation.

## RNAseq analysis shows *Salmonella*-infected tumours disrupt key genes involved in T cell metabolic reprogramming

We had established that T cells had hallmarks of defective activation both in vivo and in vitro. Therefore, we next sought to determine which signalling pathways were being disrupted by STm-infected tumours. Splenocytes were cultured with TCM (representing the TME) from infected or non-infected tumour organoids and T cells were polyclonally activated for 4 or 16 h, representing early and late timepoints for activation and *Nr4a3* expression. Following activation, CD4 T cells were FACS sorted to high purity and assessed for TCR signal, which showed that the arrested TCR signal was beginning to accumulate at 16 h in the infected TCM condition (Fig. 4A; 6.5% TCR-arrested in NT group, 28.1% TCR-arrested in STm group). CD4 T cells were then lysed and 3' mRNA libraries were created and analysed by RNAseq QuantSeq.

Principal component analysis (PCA) showed strong separation of the four groups along PC1 and PC2 axes by both timepoint and tumour treatment (Fig. 4B). Pathway analysis revealed a higher number of differentially expressed genes (DEGs) at the 16 h timepoint compared to 4 h timepoint, suggesting a progressive

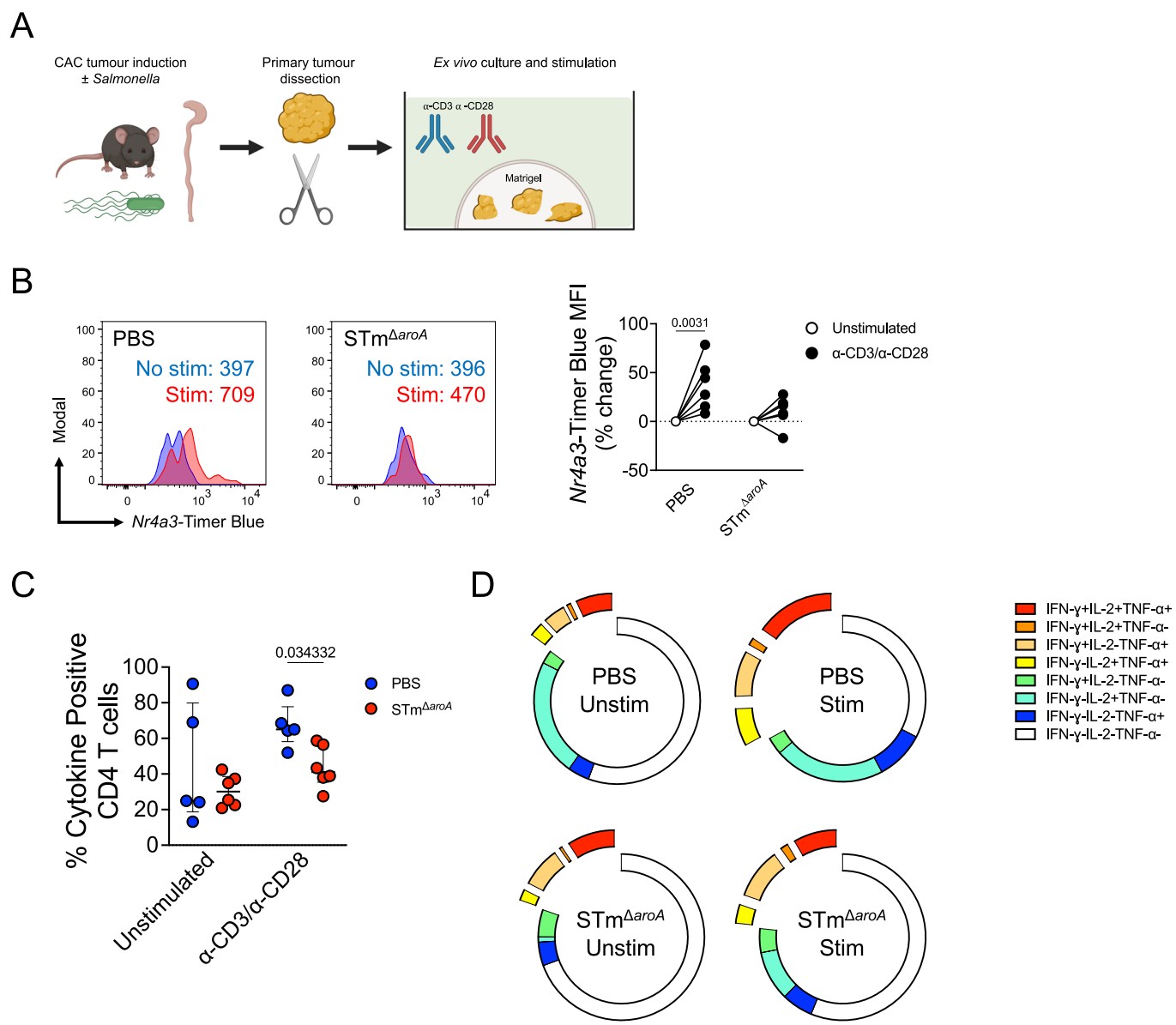

**Figure 2. TILs from mice treated with *Salmonella* are functionally defective.**

(A) Schematic depicting dissection of primary tumours in the CAC model, followed by culture in Matrigel for ex vivo synchronous activation. (B) Primary TILs were stimulated with α-CD3/α-CD28 antibodies (1 and 5 µg/mL, respectively) for 16 h, and then CD4 T cells were assessed for *Nr4a3*-Timer expression by flow cytometry, indicative of TCR-driven T cell activation. Left: Representative histograms showing T cell activation in PBS and STm-treated mice. Right: Quantified data from multiple mice showing % change in *Nr4a3*-Timer MFI upon activation. PBS $n = 6$, STm $n = 6$. (C) Tumour fragments were activated with α-CD3/α-CD28 antibodies in the presence of brefeldin A (10 µg/mL) for 6 h, and concurrent cytokine production was measured by intracellular cytokine staining (ICS; IL-2, IFNγ, TNFα). Displayed is the total cytokine production in CD4 T cells. PBS $n = 5$, STm $n = 6$. (D) Boolean analysis was performed on each possible cytokine combination in all four conditions. Shown are combined results from all mice depicting averages. Each point represents one mouse. Bars depict the median average with interquartile range. Significance was tested by two-way ANOVA with Sidak's post-test (unstimulated vs stimulated) (B) or by multiple Mann–Whitney tests with Holm–Sidak correction for multiple tests (C). B, C are from independent experiments.

reprogramming of T cell phenotype (Fig. 4C). Interestingly, the dominant pathways altered in the STm group were involved in (i) innate antiviral responses (*Stat1, Stat2, Irf7, Irf9*) and (ii) metabolic reprogramming, namely glycolysis and oxidative phosphorylation (OXPHOS). We reasoned that TLR activation of T cells was likely responsible for the innate-like signature being enriched by exposure to STm-infected tumours, and therefore we turned our attention to the metabolic pathways. Sub-analysis of metabolism-related DEGs

revealed a striking suppression of multiple genes involved in glycolysis (*Hk2, Pfkl, Pkm, Ldha, Eno1*) by *Salmonella*-infected tumours at 16 h; this was paralleled by dampening of gene expression in the OXPHOS pathway (*Cox5a, Atp5g1, mt-Nd1*) at the same timepoint (Fig. 4D). T cells exposed to non-infected tumours demonstrated normal metabolic reprogramming during this late-stage activation, with substantial upregulation of glycolysis and OXPHOS genes.

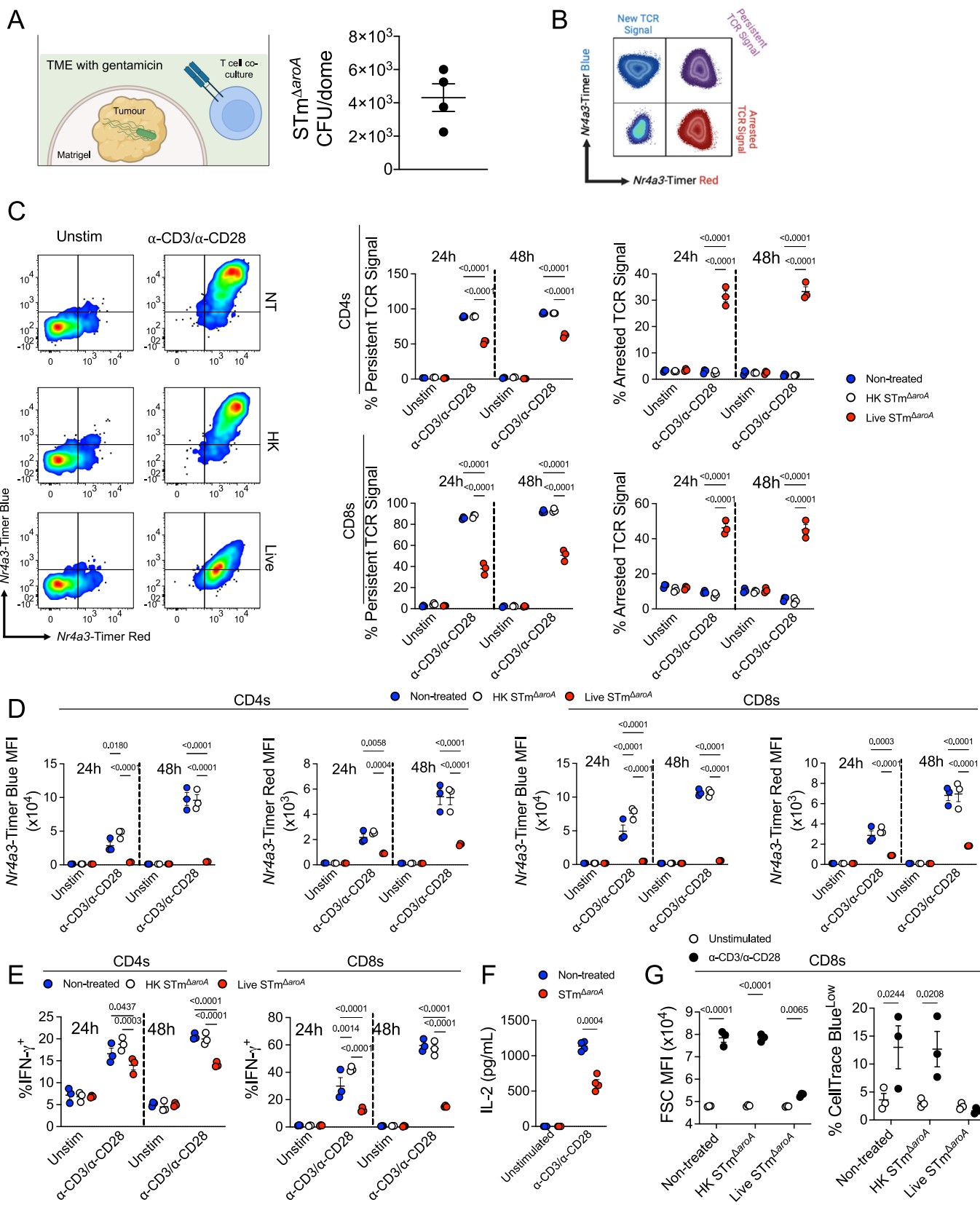

◄ **Figure 3. *Salmonella*-infected organoids recapitulate induction of T cell dysfunction and demonstrate arrested TCR activation.**

(A) Left: Schematic depicting the co-culture model of tumour organoid culture with T cells/splenocytes. Cells (1–2 × 10⁶ splenocytes) were either directly cultured with the tumour or with TCM. In direct cultures, T cells infiltrate the Matrigel. Right: Representative intra-tumoural STm CFU counts after 48 h culture, each dot represents one Matrigel dome in a 24-well plate, n = 4. (B) Schematic of flow cytometric analysis of *Nr4a3-timer* based on Bending et al, 2018a. (C) Splenocytes were cultured directly with uninfected, HK STm-treated or live STm-infected tumours for up to 48 h ± α-CD3/α-CD28 antibodies (1 and 5 μg/mL, respectively), and at each timepoint, cells were aspirated for analysis by flow cytometry at the indicated timepoints. Left: Representative plot depicting *Nr4a3*-Timer expression after stimulation in the three groups (Blue⁺Red⁺ = New TCR Signal, Blue⁺Red⁺ = Persistent TCR Signal, Blue⁻Red⁺ = Arrested TCR Signal). Right: Summary data showing the locus of TCR reporter in CD4⁺ (top) or CD8⁺ (bottom) cells. (D) Summary analysis of *Nr4a3*-Timer Blue and Red MFI values. (E) Summary data for IFN-γ⁺ T cells at 24 or 48 h. (F) Splenocytes were stimulated for 24 h in TCM from NT or STm-infected tumours, and IL-2 was measured in supernatants by ELISA. (G) Splenocytes were stained with CellTrace Blue and stimulated for 3 days in TCM from NT or uninfected tumours as described above. Left: Flow cytometric measurement of cell size (FSC-A) after stimulation. Right: Analysis of proliferating cells by gating on the CellTraceBlue^low fraction (as a percentage of CD8⁺ cells). Bars depict means ± SEM. Each point represents one mouse or organoid infection. Statistical significance was tested by two-way ANOVA comparing conditions unstimulated or stimulated groups with Tukey's post-test (C–G) or by two ANOVA with Sidak post-test comparing unstimulated to stimulated responses (F). Data were from n = 3 mice in three organoid infections and representative of more than five experiments (C–E, G), or n = 3 mice testing TCM from three independent experiments (F). Source data are available online for this figure.

## T cells exposed to *Salmonella*-infected tumours have impaired metabolic capacity

Since RNA transcriptome analysis showed several enzymes and complexes in glycolysis and OXPHOS were suppressed by STm, we next focused on the metabolic features of T cell activation when cells were exposed to infected tumours.

First, it was tested whether inhibition of glycolysis could recapitulate the antagonism of TCR signalling by STm-treated tumours, i.e. *Nr4a3*-Timer arrest (Timer Red accumulation). To this end, T cells were treated with TCM from uninfected (non-treated; NT) or STm-treated tumours containing up to 20 mM 2-deoxy-D-glucose (2-DG), a potent inhibitor of glycolysis which leads to accumulation of 2-deoxy-d-glucose-6-phosphate (2-DG-6P). As predicted, the addition of 2-DG to TCM was able to potently arrest T cell activation, as evidenced by a dose-dependent increase in arrested TCR signals in CD4s and CD8s when cultured in NT TCM (Fig. 5A). At the highest dose of 2-DG, T cells in the NT tumour group were virtually identical to the STm tumour group in terms of arresting of the TCR reporter. The addition of 2-DG to the TCM from infected tumours was able to enhance the arrested TCR signalling even further ($p < 0.01$, CD4 and CD8 stimulated: vehicle vs 20 mM 2-DG). Similar results were found for IFN-γ expression: in CD8s, 2-DG suppressed cytokine production in the NT tumour group to levels seen in the STm tumour group (Fig. 5B).

Next, splenocytes were cultured in TCM from NT or STm-infected tumours and T cell activation was induced for 24 h, followed by quantification of glucose utilisation in the culture (Fig. 5C). In cells cultured with TCM from NT tumours, there was an approximate 50% reduction in glucose ($p < 0.01$) upon T cell activation, as would be expected. In cells cultured with TCM from STm-infected tumours, however, there was no detectable glucose utilisation, consistent with the previous observation of impaired glycolytic enzyme expression. STm treatment of organoids itself depletes glucose (Mackie et al, 2021), however the levels remaining (~4 mM, Fig. 5C) should be sufficient for activation; supplementation of excess glucose could not restore the phenotype (Fig. EV2A). Since Glut1 and Glut3 are the major glucose transporters in T cells and are rapidly upregulated following activation (Macintyre et al, 2014), we assessed their expression following activation in TCM from infected or uninfected tumours. As shown in Fig. 5D, there was a severe blunting of the upregulation of Glut1/Glut3 in

response to STm, whereas T cells cultured in normal tumour media were able to strongly upregulate these glucose transporters.

To functionally explore these observations, we next performed an analysis of glycolysis and OXPHOS by measuring the extracellular acidification rate (ECAR) and oxygen consumption rate (OCR), respectively by extracellular flux analysis. As shown in Fig. 5E, T cells activated in media from STm-infected tumours demonstrated severely blunted ECAR when glucose was provided ($p < 0.0001$ NT stim vs STm stim), indicating diminished glycolytic capacity. Furthermore, following injection with electron transport chain (ETC) inhibitors, a strong trend for impaired overall glycolytic capacity and glycolytic reserve were also observed. Similar results were found for OXPHOS; the STm group demonstrated decreased maximal OCR upon injection with the mitochondrial uncoupler BAM-15, when compared to the NT group ($p < 0.05$ NT stim vs STm stim) (Fig. 5F). Strong trends were also observed for STm-mediated decreases in basal respiration, ATP-coupled respiration and spare respiratory capacity. Together with the RNAseq analysis, these results show STm induces significant impairment of T cell metabolic reprogramming.

## Activated T cells exposed to *Salmonella*-infected tumours have an intact TCR signalosome but impaired levels of c-Myc

The relationship between TCR signalling and T cell metabolic reprogramming is bi-directional: (i) "top-down", in that TCR signalling drives metabolic reprogramming to meet the energetic demands of a fully activated lymphocyte, granting optimal effector functions, and (ii) 'bottom-up', describing how changes in metabolic effectors can modulate TCR signalling and the activation trajectory of the cell (Shyer et al, 2020). Using this framework, we asked the question: are STm-infected tumours preventing T cell metabolic reprogramming due to inhibition of TCR signalling ('top-down'), or are they affecting a metabolic effector directly, so that the T cell is unable to sustain TCR-driven activation ('bottom-up')?

First, we assessed one of the earliest phosphorylation events upon TCR triggering: Lck phosphorylation (Dutta et al, 2017). Splenocytes were cultured directly with infected or non-infected tumours for 24 h, and levels of Lck phosphorylation were detected by flow cytometry after 15 min α-CD3/α-CD28 stimulation. T cells cultured with infected tumours were able to activate this kinase to a

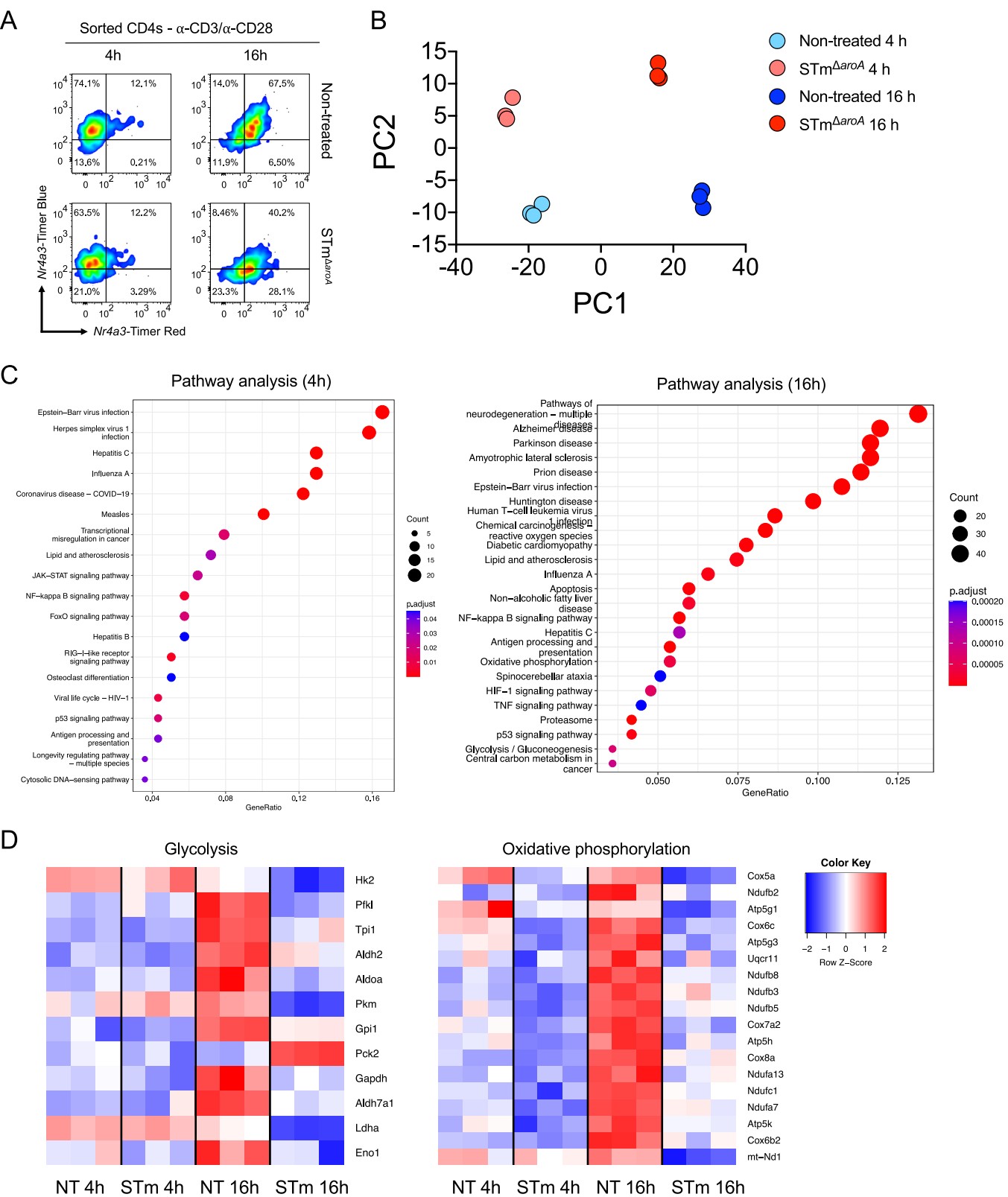

**Figure 4.  *Salmonella*-infected tumours suppress key metabolic genes in T cells, accompanied by an innate-like transcriptional signature.**

(A) Splenocytes ($1 \times 10^6$ per condition) were cultured with TCM from non-treated or STm-infected tumours for 4 or 16 h, and then purified CD4 T cells were FACS sorted from these bulk splenocytes and *Nr4a3*-Timer expression was measured by flow cytometry. The data shown are a representative mouse. (B) 3′ mRNA libraries were generated from isolated CD4s, and transcription was quantified using QuantSeq. Shown is a principal component analysis (PCA) displaying variance in the dataset along PC1 and PC2. (C) DEGs were input into the KEGG database for pathway analysis for 4 h (left) and 16 h (right). (D) Heatmap of the DEGs involved in glycolysis (left) or oxidative phosphorylation (*right*) with Z-score key. Data were derived from $n = 3$ mice, representing one of two similar experiments. All DEGs were generated with a 1.5-fold change threshold. Source data are available online for this figure.

similar extent as controls (Fig. 6A). Downstream of proximal phosphorylation events at the TCR complex is the flux of calcium that leads to NFAT translocation into the nucleus; *Nr4a3* has been established as a calcium/NFAT-dependent gene (Jennings et al, 2020), and therefore this was investigated. $Ca^{2+}$ flux activity was measured by culturing splenocytes with tumours and then briefly pulsing with the ionophore, ionomycin. As seen in Fig. 6B, T cells cultured in all conditions were fully competent at inducing calcium flux, as determined by ratiometric measurement of two cytosolic calcium dyes ($p$ = non-significant, ionomycin treatment NT vs STm in CD4s).

While calcium flux drives NFAT-dependent T cell activation, mitogen-activated protein kinases (MAPKs) induce the second axis: the AP-1/NF-κB-dependent signalling branch (Rincon and Flavell, 1996). Erk phosphorylation was therefore measured in an activation time-course from 1 to 16 h. As shown in Fig. 6C, T cells exposed to TCM from both infected and non-infected tumours could induce robust Erk phosphorylation at 1 h, which declined by 16 h. With calcium flux and Erk phosphorylation intact, we next interrogated the immediate downstream transcription factors: NF-κB and NFAT. Purified bulk T cells were cultured in TCM from NT and STm-treated tumours, and then activated for 1 h before staining for nuclear NFAT1 and NF-κB, according to a previously established protocol (Gallagher et al, 2021, 2018). In keeping with the upstream effectors, both NFAT1 and NF-κB were able to undergo equivalent nuclear translocation in both conditions, with ~20% of T cells being NFAT1$^+$ and 80% being NF-κB$^+$ irrespective of tumour infection status (Fig. 6D).

Aside from the NF-κB and NFAT1 axes, signalling via PI3K-Akt-mTOR is also critical for full metabolic reprogramming and immune functionality (Pollizzi and Powell, 2015). Hence, the activation status of this signalling pathway was measured. Surprisingly, three major proteins in this axis were similarly activated in T cells cultured with TCM from uninfected or infected tumours (Figs. 6E and EV2B–D). In fact, there was a significant increase in mTOR activation at 16 h in the STm group compared to the NT group in both CD4s and CD8s (NT: ~65% p-mTOR$^+$, STm: ~80% p-mTOR$^+$ in CD4s; Fig. EV2C). The heightened mTOR activation in the STm group most likely reflects TLR ligands driving increased activation of this pathway.

We next investigated intracellular levels of c-Myc protein, since c-Myc is a master controller of cellular metabolism (Wang et al, 2011). Unexpectedly, while T cells activated in TCM from infected tumours could upregulate c-Myc to a certain extent, this was substantially dampened compared to the NT control ($p < 0.05$ NT vs STm in CD4s and CD8s) at late-stage activation (Fig. 6F), coinciding with the observation of impaired expression of glycolytic enzymes at the same timepoint. c-Myc is a master controller of protein translation, as it governs ribosome biogenesis (van Riggelen

et al, 2010), and so we used a puromycin detection system to assess the global translational landscape. In a time-course of activation, we observed that c-Myc upregulation was rapidly suppressed in T cells exposed to TCM from infected tumours relative to the non-treated control (~40% c-Myc suppression CD4s and CD8s at 2 h, $p < 0.05$), preceding a more gradual loss of protein translation (Fig. 6G). This suggests that c-Myc is affected initially, which then has an impact on the translational capacity of the cells, explaining the failure to upregulate key glycolytic and oxidative phosphorylation enzymes. Lastly, since c-Myc is a regulator of T cell memory formation (Haque et al, 2016; Nozais et al, 2021), we also looked in vivo at the memory phenotype of activated IFN-γ$^+$ T cells in lymphatic tissue. Central memory T cells (T$_{CM}$) are much more potent than effector memory T cells (T$_{EM}$) in tumour control, and this ratio is an important biomarker for cancer survival (Liu et al, 2020). We detected an increase in T$_{EM}$-like CD4 T cells in the spleen and mLN, and a corresponding decrease in T$_{CM}$-like cells in STm-treated mice (Fig. EV3A,B, respectively), in keeping with T cell dysfunction. Taken together, these data revealed a "bottom-up" defect in T cell activation induced by STm: an intact TCR signalosome, but defective c-Myc expression leading to an inability to sustain canonical T cell activation.

## Removal of tumoural asparagine depletion by *Salmonella* restores T cell function but reverses direct control of cancer stem cells

Thus far, we had established that STm-infected tumours were suppressing T cell function by highly selective targeting of the metabolic controller, c-Myc. The next question we asked was, what is causing c-Myc suppression? c-Myc is a highly desirable drug target for tumours, given its role in promoting cancer stem cells, cell growth and proliferation. Clinical use of *E. coli*-derived L-asparaginase to deplete asparagine has a suppressive effect on c-Myc (Soncini et al, 2020), and like *E. coli*, *Salmonella* also encode asparaginases (Torres et al, 2016). Asparagine is also critical for optimal T cell function (Hope et al, 2021). To this end, we deleted *ansB*, which encodes a type II L-asparaginase, on the attenuated STm$^{\Delta aroA}$ background and used this strain to infect tumours both in vivo and in vitro (Fig. 7A–D). The asparaginase-deficient strain demonstrated equivalent infectious burden both in vitro (organoids) and in vivo (colonic tumours) (Fig. 7B,C).

Remarkably, infection of tumour organoids with STm$^{\Delta aroA/\Delta ansB}$ was able to completely reverse the arrest of TCR signalling when T cells were activated in their TME, with a near-complete reversion to persistent TCR signalling to comparable levels as those in the NT group (Fig. 7E). The bolstered TCR signalling was also reflected in IFN-γ production by these cells, with CD4 T cells gaining a significant boost compared to the STm$^{\Delta aroA}$ control ($p < 0.01$

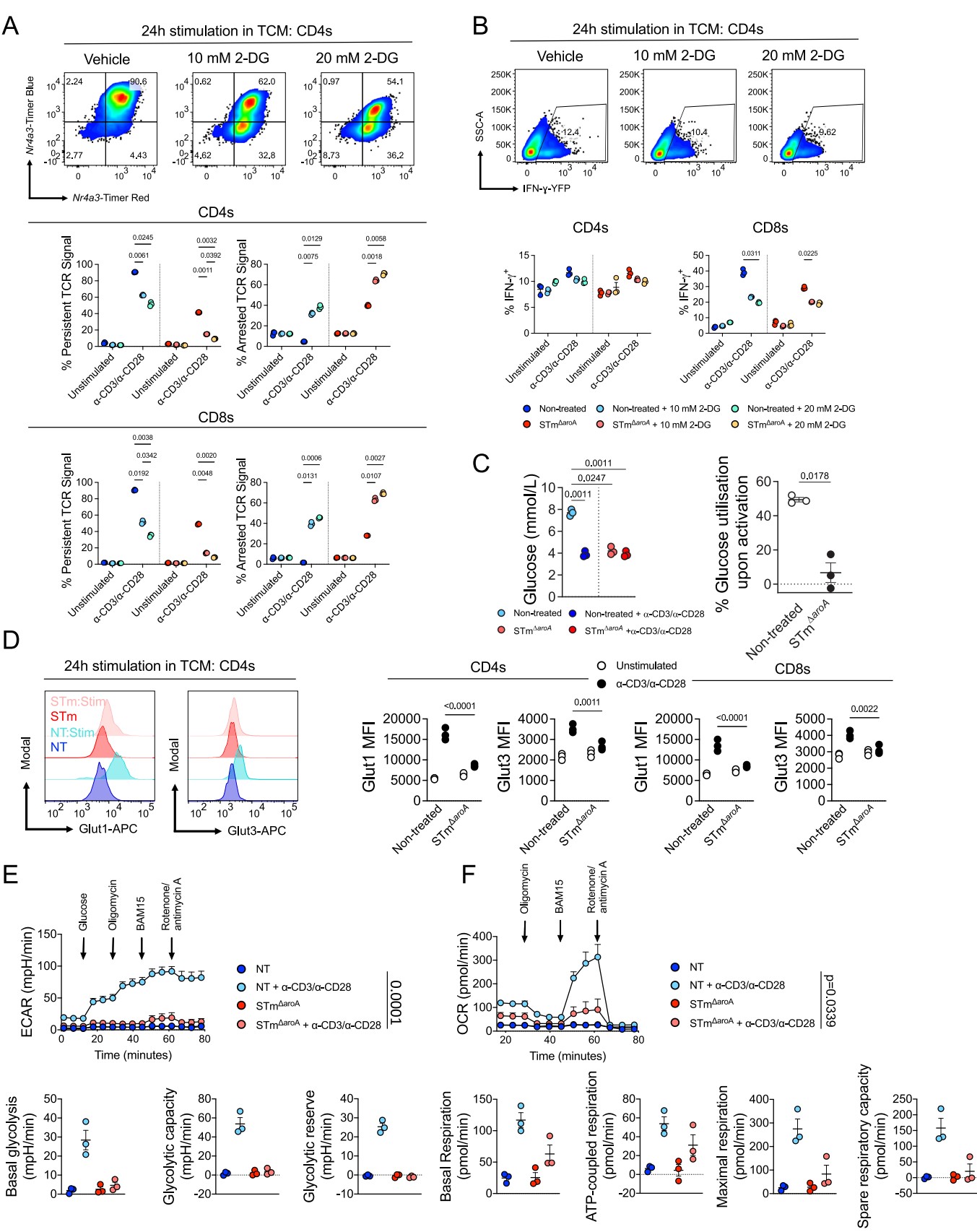

Figure 5. T cell metabolic paralysis is induced by *Salmonella*-infected tumours.

(A, B) Splenocytes (1 × 10⁶/well) were cultured with TCM from non-treated or STm-infected tumours, alongside 10–20 mM 2-DG, or vehicle control (water) added at the beginning of the experiment. Indicated groups were then activated with α-CD3/α-CD28 antibodies (1 and 5 μg/mL, respectively) and assessed for *Nr4a3*-Timer loci (A) and IFN-γ expression (B) in CD4⁺ and CD8⁺ T cells. Blue⁺Red⁺ = Persistent TCR Signal, Blue⁻Red⁺ = Arrested TCR Signal. (C) Splenocytes were cultured and activated for 24 h in TCM from NT or STm-treated tumours. Supernatants were then tested (left) for glucose concentration (Sinocare), and percent utilisation was calculated (right). (D) Splenocytes were cultured and activated for 24 h in TCM from NT or STm-treated tumours. After 24 h, cells were fixed and permeabilised. Cells were stained with either rabbit anti-Glut1 or anti-Glut3, followed by anti-rabbit-APC to reveal total Glut1/3 levels. Left: Representative histogram showing flow cytometric levels of Glut1 and Glut3 gated on CD4 T cells. Right: Compiled results from n = 3 mice. (E, F) Splenocytes were cultured in TCM from NT or STm-treated tumours for 48 h and then analysed on Agilent Seahorse XF analyser for ECAR (E) and OCR (F). Data were representative of two (A, B) or one (C) independent experiments testing n = 3 mice splenocytes on TCM from two infections, or one experiment testing TCM from two independent infections on n = 3 mice (D–F). Statistical significance was tested by two-way ANOVA comparing all groups with Sidak's post-test, displaying the stimulated in-group comparisons (A, B), or one-way ANOVA with Tukey's post-test (C), two-way ANOVA with Sidak's post-test between unstim vs stim (D) and one-way ANOVA with Tukey's post-test, displaying the stimulated group only (E, F). Source data are available online for this figure.

percentage and MFI) (Fig. 7F). To confirm that this restorative effect was due to asparagine, TCM from STm^{ΔaroA}-infected tumours was spiked with asparagine at various concentrations. Consistent with the asparaginase-deficient mutant, asparagine supplementation was able to reverse the suppression of TCR signalling and IFN-γ production by STm (Fig. EV4), alongside the restoration of c-Myc expression (Fig. 7G) to normal levels. Aspartic acid or glutamine were unable to restore T cell activation (Fig. EV4). T cells exposed to STm^{ΔaroA/ΔansB}-infected organoids also exhibited restored c-Myc expression (Fig. 7H), confirming this mechanism of dysfunction.

Asparagine is a non-essential amino acid that can be synthesised from aspartic acid precursors by asparagine synthetase (*Asns*), and so we questioned whether this was being similarly inhibited by c-Myc suppression. c-Myc can cooperate with transcription factors such as ATF4 (Tameire et al, 2019), which is critical for *Asns* expression (Krall et al, 2021). Indeed, pharmacological c-Myc inhibition vastly depletes *Asns* gene activity (Demma et al, 2019). In keeping with these observations, we detected suppressed intracellular *Asns* expression in T cells activated in TCM from STm-infected tumours during late-stage activation (48 h; Fig. 7I), thus explaining why T cells could not simply synthesise endogenous asparagine.

To further validate that asparaginase was the causative agent of T cell dysfunction, and to determine whether this effect occurs broadly amongst bacterial vectors, we tested an attenuated *Listeria monocytogenes* (*Lm*) and *E. coli* Nissle (*EcN*) vector in our tumour organoid model. *E. coli* encodes an asparaginase (*ansB*) with high homology to *Salmonellae* (92.8%), whereas *Lm* does not. Both *Lm* and *EcN* were able to invade tumour organoids (Fig. EV5A,C), albeit at lower levels than STm (see Fig. 7B for example). T cells exposed to TCM from *Lm*-infected tumours were fully functional, as evidenced by normal *Nr4a3*-Timer and IFN-γ expression (Fig. EV5B). As predicted, T cells cultured with TCM from *EcN*-infected tumours were dysfunctional compared to the control (Fig. EV5D), similar to that seen for STm^{ΔaroA}.

Since L-asparaginase therapy is known to affect tumour c-Myc (Soncini et al, 2020), we questioned whether asparagine depletion by STm may be driving some direct anti-tumour effects via depleting c-Myc. We had previously reported that STm^{ΔaroA} imposed broad metabolic changes to the tumour microenvironment as well as specific stem-modulating effects by undetermined mechanisms (Mackie et al, 2021). Thus, we now hypothesised that the same mechanism suppressing T cell function (c-Myc depletion)

was important for direct control of the tumour by *Salmonella*. To explore this, we first measured c-Myc levels in tumour organoids and found that c-Myc was indeed lowered by STm^{ΔaroA}, which could be reverted by the asparaginase-deficient STm mutant (Fig. 8A). Using an MTT assay to quantify the metabolic activity of the tumours and microscopy to assess organoid growth qualitatively, we found that STm^{ΔaroA} infection significantly reduced the metabolic activity of tumour organoids (*p* < 0.001 NT vs STm^{ΔaroA}), with the appearance of darkened, blebbing organoids (indicating cell death), which was reversed by addition of exogenous asparagine or the asparaginase-deficient mutant (Fig. 8B,C). To link this to c-Myc, we also then tested a small molecule inhibitor to see if this could replicate the effects of STm^{ΔaroA} infection, which was found to be the case (Fig. 8D). We next assessed the effect of excess asparagine or asparaginase-deficient STm on tumour organoid expression of the stem cell marker *Lgr5*, as we have previously reported that STm^{ΔaroA} causes a reduction of multiple indications of tumour stemness, including expression of *Lgr5*. There was a trend for a reduction in *Lgr5* expression by STm^{ΔaroA}, which was abrogated by asparagine or the asparaginase mutant (Fig. 8E). This phenotype could also be replicated by the addition of the c-Myc inhibitor, which potently suppressed tumour stemness in a dose-dependent manner (Fig. 8F). Intracellular analysis of Asns protein revealed equivalent levels between NT and STm-treated tumour organoids (Fig. 8G). To test whether asparaginase could directly antagonise tumour growth, we added *E. coli*-derived asparaginase to tumours and measured growth by MTT, showing a dose-dependent halting of tumour metabolism (Fig. 8H), indicating that intestinal tumour organoids are sensitive to asparagine depletion.

Given that STm asparaginase could also antagonise tumour c-Myc expression, effecting organoid growth and *Lgr5* expression, we now questioned how protective the STm^{ΔaroA/ΔansB} double mutant would be in vivo. On one hand, loss of tumour intrinsic c-Myc suppression might be disadvantageous, but on the other, gain of T cell functionality may impart a net benefit. We treated AOM-DSS CAC-induced mice with PBS, STm^{ΔaroA} or STm^{ΔaroA/ΔansB} weekly for 6 weeks. As previously reported STm^{ΔaroA}-treated mice have reduced tumour number and size (Fig. 8I). We found that *Salmonella* additionally lacking asparaginase also had a significant reduction in tumour load, indicating that STm^{ΔaroA/ΔansB} is still capable of reducing tumour growth in vivo (Fig. 8I). As previously described, tumours from STm^{ΔaroA} treated mice had reduced expression of

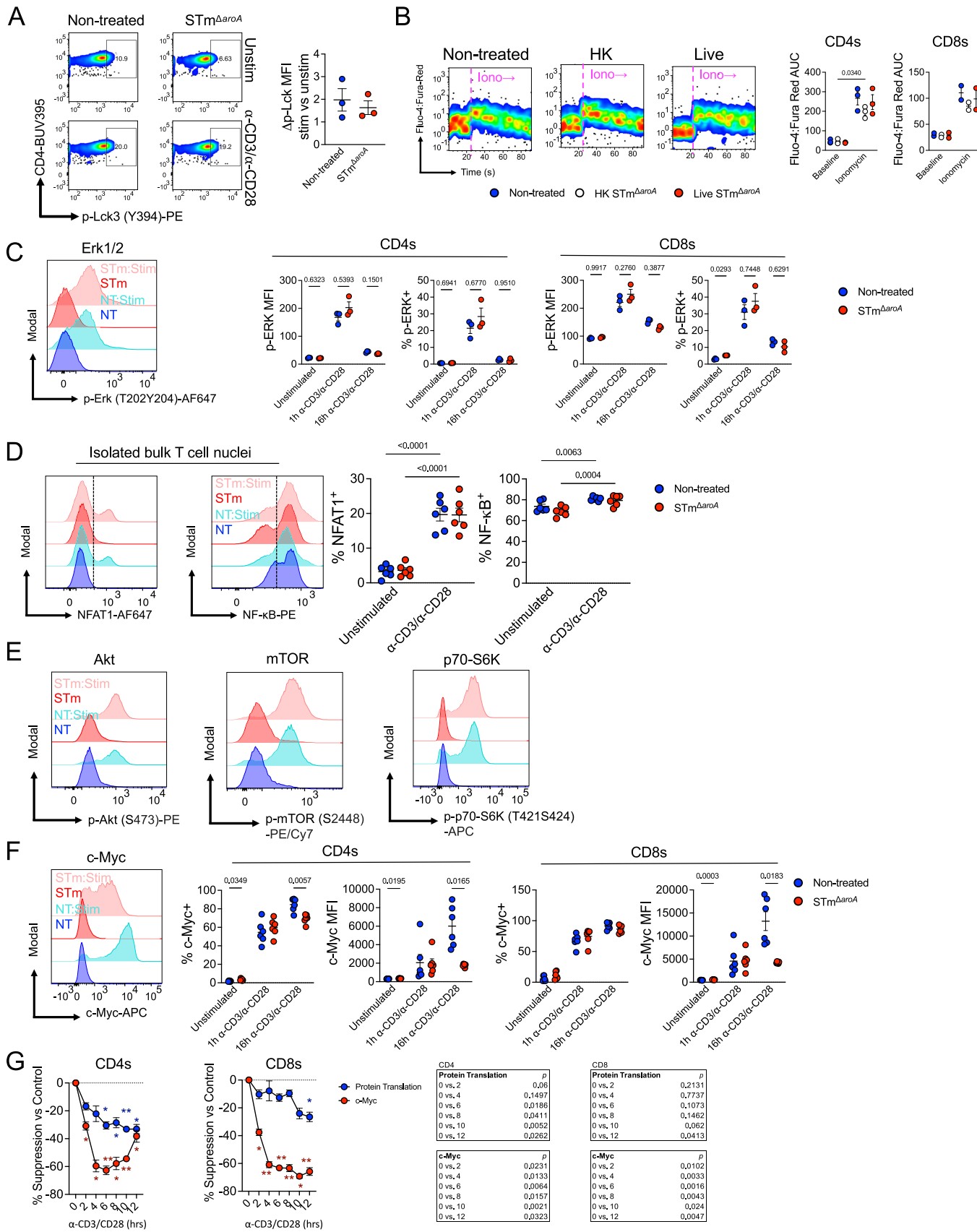

◄  **Figure 6.  c-Myc is selectively inhibited by STm-treated tumours, but the TCR signaolsome remains intact.**

(A) Splenocytes were cultured directly with non-treated, HK STm-treated or live STm-infected tumours for 24 h. Cells were then stimulated for 15 min with α-CD3/α-CD28 antibodies (1 and 5 μg/mL, respectively) and then stained, fixed and permeabilised according to the PhosFlow protocol (see Methods). Cells were stained for Lck phosphorylation (Y394) and analysed by flow cytometry. Data shown are gated on $CD4^+$ T cells. $n = 3$ all groups. (B) Splenocytes were cultured directly with NT, HK STm-treated or live STm-infected tumours for 24 h. Cells were then stained with lineage markers and Fluo-4/Fura-Red (calcium dyes), and run on a flow cytometer to measure baseline calcium. Ionomycin (1 μg/mL) was added after ~25 s, and calcium flux was measured by a ratio of Fluo-4:Fura-red and calculating the area under curve (AUC) before and after treatment. Left: Representative plots showing calcium flux for each group, gated on CD4 T cells. Right: Pooled data from $n = 3$ (CD4) or $n = 2$ (CD8) mice. (C) Splenocytes were activated in TCM for 1 or 16 h with α-CD3/α-CD28. Cells were stained according to the PhosFlow protocol, and p-Erk (T202/Y204) was measured. Left: representative histogram showing induction in both activated groups. Right: Pooled data from $n = 3$ mice, all groups. (D) Purified bulk T cells (>98% purity) were cultured in NT or STm TCM for 24 h. One hour before the end of the culture, a-CD3/a-CD28 was added to the indicated groups to measure nuclear translocation. Nuclei were then separated and analysed by flow cytometry for NFAT1 and NF-kB. Left: Representative histograms showing NFAT1 and NF-kB translocation after activation. Right: Pooled data from three experiments, $n = 6$ mice, all groups. (E) Representative flow cytometry histograms of p-Akt (S473), p-mTOR (S2448) and p-p70-S6K (T421/S424). Cells were stimulated in NT and STm TCM for 1 h and processed as previously described. Data representative of $n = 3$ mice. Quantification shown in EV Fig. 2B–D. (F) Splenocytes were cultured in TCM for 1 or 16 h with a-CD3/a-CD28. Cells were stained, fixed, permeabilised and stained with rabbit anti-c-Myc, followed by anti-rabbit APC. *Left*: Representative histogram after 1 h stimulation. *Right*: Data pooled from two experiments, $n = 6$ mice, all groups. (G) Splenocytes were activated in TCM from non-treated or STm-infected tumours with α-CD3/α-CD28 for up to 12 h, and global protein translation was measured by intracellular puromycin staining, alongside a separate c-Myc stain. Differences in %ΔMFI for puromycin (protein translation) or c-Myc in the STm group vs the non-treated control at each timepoint are shown. $n = 3$ all groups (but $n = 2$ 10 h c-Myc). Data shown are pooled from two (B, F), three (D, G) or one (E) independent experiments, or representative of two/three independent experiments (A, C, respectively). Bars depict means ± SEM. Statistical significance was tested by two-way ANOVA with Sidak post-test between treatments (B, D), timepoints (NT vs STm) (C, F) and mixed-effects model with Dunnett's post-test (G).

tumour stem cell markers *Lgr5* and *Smoc2*, with a strong trend in *Vim*, a mesenchymal marker (Mackie et al, 2021), which is a desirable outcome for tumour control (Fig. 8J). Strikingly, oral treatment with $STm^{\Delta aroA/\Delta ansB}$ negated this intrinsic tumour control, as evidenced by expression of *Lgr5*, *Smoc2*, and *Vim*, to levels comparable to the PBS control group, in contrast to the $STm^{\Delta aroA}$ treatment, which reduces expression of these genes.

Finally, to explore the restoration of T cell functionality further, we assessed IL-2 production in splenocytes cultured with $STm^{\Delta aroA}$ or $STm^{\Delta aroA/\Delta ansB}$-infected tumours, as well as primary colonic tumour fragments cultured in Matrigel, taken from mice that had received oral gavages of either strain of bacteria (as described in Fig. 2A). As shown in Fig. 8K, asparaginase deletion was able to restore, or even enhance, IL-2 production in our co-culture model and within primary tumours.

In summary, while restoring T cell function using an asparaginase-deficient STm comes at the cost of some tumour cell intrinsic control via c-Myc, there is still sustained tumour reduction and so T cell functionality alongside other tumour control mechanisms of STm elicit significant protection from autochthonous tumour growth.

## Discussion

Bacterial cancer therapy occupies a unique niche in the cancer therapy landscape, being historically the oldest immunotherapy, yet at the forefront of biotechnological innovations. Bacterial vectors can be extensively engineered with additive functions (e.g. expression of cytokines (Yoon et al, 2017), checkpoint inhibitors (Pan et al, 2022), metabolic enzymes (Canale et al, 2021)), to great success in preclinical cancer models. To date, however, there is only one licensed BCT therapy in modern healthcare: the Bacillus Calmette–Guérin (BCG) vaccine, which is used to treat superficial bladder cancer (Pettenati and Ingersoll, 2018). Clinical trials using bacterial vectors—including *Salmonella*—have generally shown lacklustre results (Toso et al, 2002). It may be speculated that the naturally harboured immune evasion mechanisms these bacteria possess are a barrier to optimal therapeutic success, and one that is seldom explored.

Attenuated *Salmonella* is an ideal treatment vector for CRC for several reasons: (i) it matches the natural route of infection to the tumour location, and (ii) provides an opportunity for biological therapy in those patients whose cancer is poorly responsive to ICB—microsatellite instability[lo], which comprises the majority of CRC patients (André et al, 2020; Zhao et al, 2019). The immune response to STm BCT has been thoroughly investigated, revealing the requirement of innate cells and innate immune adaptors for treatment efficacy (Chen et al, 2021; Johnson et al, 2021; Zheng et al, 2017). Yet, while T cells have been shown redundant for treatment efficacy for two decades (Luo et al, 2001), canonical T cell activation and function is still assumed. This presumption is made despite the extensive immune evasion mechanisms harboured by *Salmonella* (Bernal-Bayard and Ramos-Morales, 2018; Wang et al, 2020). *Salmonella* spp. can block cellular immunity during natural infection by multiple means, including modulation of DC maturation (Aulicino et al, 2018), inhibition of MHC peptide loading (Lapaque et al, 2009), and direct LPS-mediated toxicity (Srinivasan and McSorley, 2007). While asparagine depletion is an established immune evasion mechanism by bacteria during infection (Torres et al, 2016), this is the first study to link this to T cell redundancy during BCT. Indeed, to our knowledge, no study has yet assessed the role of any bacterial immune evasion mechanism in BCT, which is an area ripe for investigation.

Using a transgenic TCR reporter mouse that included temporal analysis of TCR signalling, we showed for the first time that T cells are suppressed during BCT, exhibiting defects in sustaining TCR signalling, cytokine production and proliferation. This was not due to problems in the TCR signalosome per se, but rather, a highly specific suppression of one master controller of T cell metabolism: c-Myc. Deletion of a single bacterial enzyme, or addition of a single amino acid, was sufficient to restore c-Myc expression and canonical T cell activation. In other words, T cells were poised to respond to TCR signalling, but prevented from sustaining an activation trajectory due to asparagine depletion. This effect is likely to be dependent on the local environment, in particular the presence of live STm.

This reinvigoration, however, came at the cost of some intrinsic tumour cell control by STm. We have shown previously that STm can

directly affect intestinal tumours by suppressing their stem cell-like characteristics, and also by imposing metabolic competition (Mackie et al, 2021). The data herein provide some mechanistic underpinning for these previous observations, and a target to improve therapeutic outcomes. c-Myc is a highly desirable oncotarget but one that is challenging for drug design due to the ubiquitous and essential nature of c-Myc in both tumour and normal cells (Madden et al, 2021). *Salmonella*, as a tumour-homing vector, and one which can preferentially invade Lgr5+ cancer stem cells (Mackie et al, 2021), is uniquely placed to exploit this pathway in tumour control.

The finding of dysregulated T cells during therapy is unlikely to represent a more general exhaustion caused by infection per se, since PD-1 was found to be negatively affected by bacterial infection. Indeed, infections which establish chronicity (such as *B. burgdorferi*, HBV or *T. gondii*) induce high levels of PD-1 as a core feature (Bhadra et al, 2012; Boni et al, 2007; Helble et al, 2022). In accordance, stimulation of primary tumours from infected mice ex vivo exhibited a reinvigorated T cell IL-2 response when asparaginase was deleted from the bacterium. This deletion had no effect on bacterial burden within the mice, thus excluding exhaustion as a causative factor. *Salmonella* itself rarely establishes chronic infections, and the attenuated strain used in the current study is yet more rapidly cleared than wild-type *Salmonella*.

One element that we did not resolve is the innate-like antiviral signature that T cells exhibited in the RNAseq analysis. However, these data are completely consistent with the later finding of diminished c-Myc protein by STm. Multiple components of the type I IFN pathway, such as IFN-β and IRF7, can antagonise c-Myc expression (Kim et al, 2016; Sarkar et al, 2006). This relationship is reciprocal, since c-Myc can, in turn, lead to degradation of key transcription factors in the type I IFN pathway, including STAT1 (Schlee et al, 2007). This early innate-like signature, therefore, is highly likely to reflect a rapid loss of lymphocyte c-Myc in the absence of asparagine.

A limitation to our study is that we did not account for the possibility of T cell migration or possible diffusion of the asparaginase molecule itself. Therefore, it is difficult to know whether T cells received any 'normal' priming in a distal anatomical site (hence explaining retained IFN-γ-YFP in vivo, but decoupled Nr4a3-Timer expression). When injected intravenously, asparaginase can accumulate in lymphatic tissues, thus it is possible that STm is capable of secreting this enzyme for far-reaching distal biological effects. Furthermore, we cannot fully exclude that the observed differences in the T cell memory compartment are due to infection per se, and not antagonism of the c-Myc pathway. Lastly, we did not investigate the contribution of Tregs to these phenomena, due to Foxp3 detection causing a loss of intracellular fluorescent timers. Tregs possess differential metabolic properties to conventional T cells, and it will be intriguing to test whether these cells are equivalently affected by asparagine depletion. These questions will be an interesting area of future investigation in terms of both BCT and asparagine-depleting therapies.

We speculate that this study could have important implications for other bacterial vectors, namely the *Escherichia coli* strain Nissle (EcN). EcN is a probiotic bacterial strain that has been studied extensively in the prevention of CRC, and has recently been used in a study to modulate the metabolome of the TME (Canale et al, 2021) and investigated for use in the detection and treatment of colorectal cancer (Gurbatri et al, 2024). EcN contains a homologue of the *Salmonella* asparaginase that paralysed T cell metabolism in the present study—in fact, *E.coli* asparaginase is used as a chemotherapeutic. Countering this anti-T cell property—while sparing the direct anti-tumour effects—could unleash the full therapeutic potential of these bacterial vectors.

As to whether we can improve the therapeutic effect by balancing c-Myc suppression in tumour cells whilst sparing T cells remains to be determined. Additionally, it will be useful to untangle whether T cells are directed against tumoural and/or bacterial antigens during treatment. These points will be the subject of further investigation by our laboratory. Another consideration is whether certain tumours might respond better to bacterial therapy with an asparaginase-deficient STm, for example tumours with less reliance on c-Myc.

In summary, we have shown for the first time that T cells are metabolically disabled in *Salmonella*-based BCT and proposed a model of asparagine depletion by the bacteria as the mechanism underpinning this. This asparagine depletion, however, is crucial for the direct anti-tumour effects of the bacteria. Anti-T cell and anti-tumour effects in BCT are, therefore, inexorably tied and must be disentangled for maximal treatment potential.

# Methods

**Reagents and tools table**

| Reagent/resource | Reference or source | Identifier or catalogue number |
|---|---|---|
| **Experimental models** | | |
| *Nr4a3*-Tocky | (Bending et al, 2018a, 2018b (m.ono@imperial.ac.uk) | na |
| Great-Smart *ifng*-YFP reporter | Price et al, 2012, https://doi.org/10.1371/journal.pone.0039750 | na |
| Nur77-Tempo | (Elliot et al, 2022); Infrafrontier | EMMA ID: EM:15078 |
| **Antibodies** | | |
| β—tubulin-AF488 (1:50) | Cell Signaling | 9F3 |
| ASNS (1:100) | Cambridge Bioscience | polyclonal |
| CD3e (1 µg/mL) | Biolegend | 145-2C1 |
| CD28 (5 µg/mL) | Prof Anne Cooke, University of Cambridge | 37.51 from hybridoma |
| CD4-BUV395 (1:200) | BD | GK1.5 |

| Reagent/resource | Reference or source | Identifier or catalogue number |
|---|---|---|
| CD4-BUV737(1:200) | BD | GK1.5 |
| CD4-BV421 (1:200) | Biolegend | GK1.5 |
| CD8a-PE/Cy7 (1:200) | Biolegend | 53-6.7 |
| CD8a-AF700 (1:200) | Biolegend | 53-6.7 |
| CD45-BUV395 (1:200) | BD | 30-F11 |
| CD45-FITC (1:200) | Biolegend | 30-F11 |
| CD45-PerCP/Cy5.5 (1:200) | Biolegend | 30-F11 |
| CD69-AF700 (1:100) | Biolegend | H1.2F3 |
| CD44-PE/Cy7 (1:100) | Biolegend | IM7 |
| CD62L-APC (1:100) | Biolegend | MEL-14 |
| TCRb-PerCP/Cy5.5 (1:100) | Biolegend | H57-597 |
| TCRb-AF700 (1:100) | Biolegend | H57-597 |
| PD-1-PE/Cy7 (1:100) | Biolegend | 29 F.1A12 |
| CD90.2-APC (1:100) | Biolegend | 53-2.1 |
| Glut1 (1:100) | Invitrogen | SA0377 |
| Glut3 (1:100) | Invitrogen | JA50-31 |
| p-Erk1/2-T202/Y204-AF647 (1:100) | Biolegend | 4B11B69 |
| NFAT1-AF647 (1:50) | Cell Signaling | D43B1 |
| NF-kB-PE (1:50) | Cell Signaling | D14E12 |
| p-Akt-S473-PE (1:100) | eBioscience | SDRNR |
| p-mTOR-S2448-PE/Cy7 (1:100) | eBioscience | MRRBY |
| p-p70-S6K-T421/S424 (1:100) | Cell Signaling | Polyclonal rabbit |
| IL-2-PE (1:50) | Biolegend | JES6-5H4 |
| puromycin-AF647 (1:200) | Merck | 12D10 |
| TNFa-BV605 (1:50) | Biolegend | MP6-XT22 |
| IFN-g-AF647 (1:50) | Biolegend | XMG1.2 |
| c-Myc (1:100) | Cell Signaling | E5Q6W |
| p-Lck-Y394 (1:100) | Biolegend | A18002D |
| F(ab')2-Goat anti-Rabbit IgG(H + L) Cross-Adsorbed-APC (1:500) | Invitrogen | F-2765 |
| eFluor-780 Fixable Viability Dye (1:1000) | Invitrogen | 65-0865-14 |
| **Oligonucleotides and other sequence-based reagents** | | |
| Lgr5 5′ | PeproTech® | CGGAAAGTGGAATCCTTGCA |
| Lgr5 3′ | PeproTech® | CACATCGATCTGGACATGCTGT |
| Vim 5′ | PeproTech® | CGGAAAGTGGAATCCTTGCA |
| Vim 3′ | PeproTech® | CACATCGATCTGGACATGCTGT |
| Smoc2 5′ | PeproTech® | CCCTCAGAAGCCACTCTGTG |
| Smoc2 3′ | PeproTech® | ACTTGCTGGAACTCCTTCCG |
| Gapdh 5′ | PeproTech® | TGTGTCCGTCGTGGATCTGA |
| Gapdh 3′ | PeproTech® | TTGCTGTTGAAGTCGCAGGAG |
| **Chemicals, enzymes, and other reagents** | | |
| Matrigel | Corning® | **CLS356231** |
| EGF | PeproTech® | AF-315-09 |
| Cell Recovery solution | Corning® | CLS354253-1EA |
| MTT | Abcam | ab146345 |

| Reagent/resource | Reference or source | Identifier or catalogue number |
|---|---|---|
| DMEM-F12 | Gibco | 12634028 |
| Penicillin | Gibco | 15140122 |
| Streptomycin | Gibco | 15140122 |
| gentamicin | Gibco | 15750037 |
| HEPES | Gibco | 15630080 |
| Glutamax | Gibco | 35050061 |
| N-2 supplement | Gibco | 17502001 |
| B27 supplement | Gibco | 17504001 |
| N-acetylcysteine | Merck | A9165 |
| Oligomycin 1 μM | Merck | 495455 |
| BAM-15 3 μM | Merck | SML1760 |
| Rotenone 2 μM | Merck | 557368 |
| Antimycin A 2 μM | Merck | A8674 |
| $MgCl_2$ | Merck | 208337 |
| Sucrose | Merck | S0389 |
| EDTA-free protease inhibitor | Roche | 11836170001 |
| Triton X-100 | Merck | T8787 |
| Fura-Red | Invitrogen | F3020 |
| Fluo-4 | Invitrogen | F14201 |
| Fixation buffer | Biolegend | 420801 |
| TruePhos fixation buffer | Biolegend | 425401 |
| eBioscience Foxp3/Transcription Factor Staining Buffer Set | Invitrogen | 00-5521-00 |
| EDTA | Merck | E6758 |
| IL-2 Biolegend ELISA Max Deluxe Kit | Biolegend | 431004 |
| Cell Trace Blue | Invitrogen | C34574 |
| c-Myc inhibitor | SelleckBiochem | 10058-F4 |
| Asparaginase | 2B Scientific Ltd | RPPB1977 |
| L-Glutamine | Merck | G8540 |
| Asparagine | Sigma-Aldrich | A4159 |
| Aspartic acid | Merck | A7219 |
| D-glucose | Merck | G7021 |
| ACK Lysis Buffer | Invitrogen | A1049201 |
| Brefeldin A | Merck | B5936 |
| β-Mercaptoethanol | Merck | M3148 |
| DNase I | Roche | 10104159001 |
| Collagenase D | Roche | COLLD-RO |
| azoxymethane | Merck | A5486 |
| Dextran sodium sulfate (DSS) $M_W$35-50,000 | APExBIO | B8205-APE |
| LB (Lennox) | Merck | L3022 |
| BHI agar | Merck | 70138 |
| Qiagen RNeasy Mini Kit | | |
| Seahorse XF serum-free RPMI | Agilent | 103681-100 |
| pre-coated (poly-D-lysine) Seahorse plates | Agilent | 103799-100 |
| Arcturus Picopure RNA kit | ThermoFisher | KIT0204 |

| Reagent/resource | Reference or source | Identifier or catalogue number |
|---|---|---|
| **Software** | | |
| FlowJo version 9 | BD | |
| Prism version 9 | GraphPad | |
| BlueBee QuantSeq | Lexogen | |
| **Other** | | |
| LightCycler 480 | Roche | |
| Seahorse XFe96 metabolic extracellular flux analyser | Agilent | |

## Mice

Three strains of mice were used in this study: (i) C57BL/6 (Charles River, UK), (ii) *Nr4a3*-Timer ('Tocky') (Bending et al, 2018a, 2018b) crossed with Great-Smart *ifng*-YFP reporter (Price et al, 2012) and (iii) Nur77-Tempo mice (Elliot et al, 2022) (EMMA ID: EM:15078). Males and females aged 6–32 weeks were used throughout the experiments, with age matching wherever possible. All animal experiments were approved by the Institutional Animal Care and Use Committees of RIKEN Yokohama Branch (計2018-1(3)) and Yokohama City University (T-A-17-001) (Fig. 8I,J) or the University of Birmingham local animal welfare and ethical review body and authorised under the authority of Home Office license P06118734 (held by K.M.M.) (all other experiments). Animals were maintained under SPF conditions with 12 h day/night cycles and chow and water were fed ad libitum. Cages were supplemented with enrichments such as chew sticks, tubes for handling and domed houses. The ARRIVE guidelines were followed as applicable.

For use of *Nr4a3*-Tocky mice, please contact Dr Masahiro Ono (m.ono@imperial.ac.uk).

## Bacteria

*Salmonella enterica* serovar Typhimurium (SL3261) with a deletion in *aroA* (3-phosphoshikimate 1-carboxyvinyltransferase) was kindly provided by Professor Adam Cunningham (University of Birmingham, UK) (Flores-Langarica et al, 2015). The ΔansB mutant from STm 14,028 s single gene deletion library (Porwollik et al, 2014) was transduced into an *aroA* deleted STm UF20 strain (SR-11 × 3181 (Gulig and Doyle, 1993)) using P22 phage-mediated transduction (Schmieger, 1972). For experiments involving the ΔansB mutant, the UF20 parent strain was used as an isogenic control. *E.coli* Nissle 1917 (EcN) strain was kindly provided by Ardeypharm GmbH. Listeria monocytogenes (Lm) listeriolysin-deficient strain (ΔactA) (Pepper et al) was kindly provided by Professor David Withers (University of Birmingham, UK). STm and EcN strains were grown in LB broth (Merck) and Lm in BHI broth, each with relevant antibiotics, overnight to saturation, followed by 1:20 sub-culture the following day in LB or BHI until an $OD_{600}$ of 0.7 was reached, indicating logarithmic growth. This sub-culture was then washed in PBS, concentrated, and used to inoculate either tumour organoids or mice. Colony-forming units (CFU) were enumerated in tumour lysates on LB or BHI agar (Merck) using standard microbiological techniques.

## Colitis-associated colorectal cancer (CAC) model

C57BL/6 WT or *Nr4a3*-Timer-GS mice were used to induce CAC. About 7–8 week-old mice (C57BL/6 WT or *Nr4a3*-Timer-GS) were injected with 10 mg/kg azoxymethane (Merck) via i.p. on day 0. On day 1, mice were given 3% dextran sodium sulfate (DSS) $M_W$ 35–50,000 (APExBIO) in drinking water for ~7 days, followed by a 2-week rest, and then two more cycles of 3% DSS. After a 1–2 week rest from DSS, mice were split into treatment groups and were orally gavaged with $5 \times 10^9$ CFU bacteria in 100 μL volume, or PBS control, followed by a week rest, and then another dose of oral bacteria. A week later, mice were sacrificed, and tissue was collected for immunophenotyping. For tumour burden experiments, mice received 6 weekly doses of STm instead of 2, after which tumour burden was enumerated within the colon by inverted light microscopy. Treatment groups were split among cages of induced mice—using weight loss during the DSS phases as a guide, so that each treatment group had overall similar weight loss. There was no blinding of groups.

Experiments using *Nr4a3*-Timer-GS were littermate mice and both males and females used. Experiments using purchased C57BL/6 mice used only female mice.

## Tumour digestion and TIL analysis

Tumours were dissected into tumour media (TM) containing ice-cold advanced DMEM-F12 (Gibco) containing 100 U/mL penicillin, 100 μg/mL streptomycin (Merck), 2% FCS (Gibco), 25 mM HEPES (Merck) and 50 μM β-mercaptoethanol (Merck). The tissue was then mechanically dissected into ~1 mm fragments using sterile scissors. Tumour fragments were then washed in 10 mL TM, and the pelleted fragments were resuspended in pre-warmed 10 mL digestion buffer (DMEM-F12 containing 100 U/mL penicillin, 100 μg/mL streptomycin, 2% FCS, 0.1 mg/mL DNAse I (Roche) and 1 mg/mL collagenase D (Roche)) and digested in a 37 °C orbital shaker under maximum rotation for 30–45 min. Lastly, the digestion mixture was passed through a 70-μm cell strainer (BD Falcon), washed in FACS buffer, and processed for flow cytometry.

## Tumour organoid infection

Tumour organoids were derived from primary colonic tumours from the CAC model (C57BL/6 background, female) or small intestine and colonic tumours from adenomatous polyposis coli heterozygous mutant mice (Apc^min/+, male and female) (described in Mackie et al, 2021). Organoids were grown in 50 μL Matrigel (Corning) domes in 500 μL of tumour organoid media (DMEM-F12 containing 100 U/mL penicillin, 100 μg/mL streptomycin, 25 mM HEPES, 1:100 Glutamax (Gibco), 1:100 N-2 supplement (Gibco), 1:50 B27 supplement (Gibco), 1.25 mM *N*-acetylcysteine

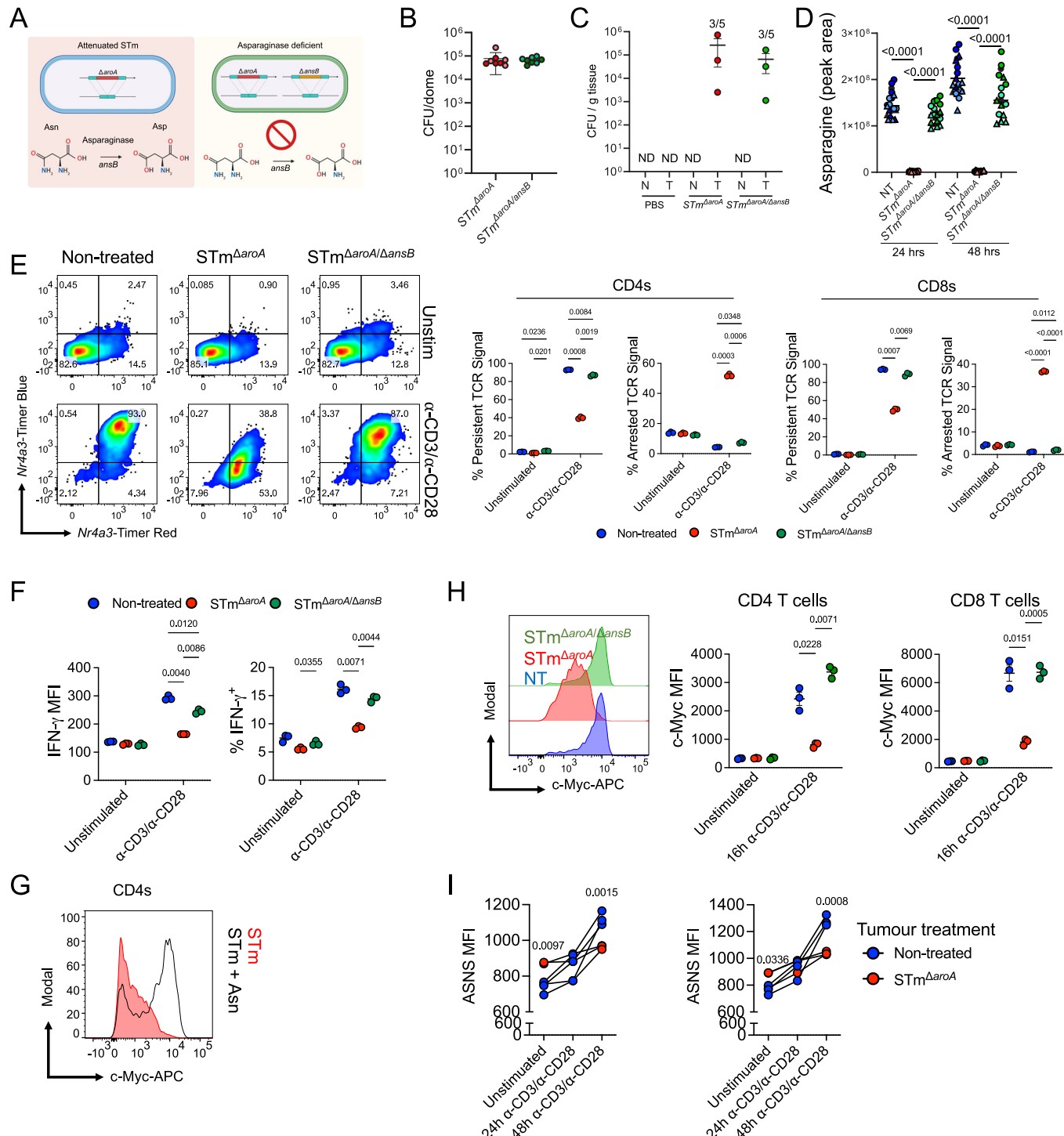

(Merck) and 50 pg/mL mEGF (Peprotech)). On day 3–4 of seeding, an overnight culture of STm was sub-cultured 1:20 in LB broth to an $OD_{600}$ of ~0.7. An inoculum of 5 µL containing ~$1 \times 10^8$ bacteria was added to each well, and then incubated at 37 °C in 5% $CO_2$ for 2 h. Next, the media was aspirated and discarded, and each well was washed with fresh PBS twice. Finally, fresh tumour organoid media (500 µL) containing 50 µg/mL gentamicin (Gibco) was added, and the organoids were cultured for 24–48 h, as indicated. Tumour organoids were routinely passaged on a weekly basis and checked microscopically for organoid-like morphology. Mycoplasma testing was conducted, and all lines were negative. Heat-killed bacteria were prepared by heating the sub-culture to 95 °C for 10 min, and

**Figure 7. Deletion of bacterial asparaginase *ansB* restores T cell function.**

(A) Schematic depicting the asparaginase-deficient attenuated STm mutant and the impaired catalytic reaction. (B) CFU analysis of organoids NT or infected with $STm^{\Delta aroA}$ or $STm^{\Delta aroA/\Delta ansB}$ for 24 h. Each dot represents an individual Matrigel dome, representative of >3 repeats in two independent organoid lines. The pale colour represents small-intestinal-derived tumour and darker shade represents colonic-derived tumour organoids, both from $Apc^{min/+}$. $n = 4$ per group. (C) CFU analysis of normal (N) or tumour (T) tissue from AOM/DSS induced mice, given two oral doses of indicated STm, once per week, and culled on 3rd week. Representative of two experiments. $n = 5$ per group. (D) In vitro measurement (LC-MS) of asparagine in culture supernatant after 24 or 48 h infection with $STm^{\Delta aroA}$ or $STm^{\Delta aroA/\Delta ansB}$, or non-treated (NT). Circles indicate colon tumour organoid, triangles small intestine; dark and light tones indicate independent experiments. One-way ANOVA with Turkey's multiple comparisons test was performed on the means (black circles/triangles) of each experiment. $n = 8$ per group. (E) Splenocytes were cultured with TCM from tumours infected with $STm^{\Delta aroA}$ or $STm^{\Delta aroA/\Delta ansB}$ and activated with α-CD3/α-CD28 for 24 h. Left: representative flow cytometric plot showing *Nr4a3*-Timer, gated on CD4 T cells. *Right*: pooled data from multiple mice, showing the Timer loci in relation to TCR signalling in CD4 and CD8 T cells. TCM pooled from two independent infections, $n = 3$ mice. (F) IFN-γ expression in the same experiment (E), showing pooled responses gated on CD4 T cells. (G) c-Myc levels in CD4 T cells after supplementation with 0.5 mM asparagine, as measured by intracellular flow cytometry. Representative of one of $n = 2$ mice. (H) Splenocytes were cultured with TCM from tumours infected with STmΔ$^{aroA}$ or STmΔ$^{aroA/ansB}$ and activated with α-CD3/α-CD28 for 16 h. c-Myc expression in CD4 and CD8 T cells, representative flow cytometric plot from CD4 T cells. $n = 3$ mice. (I) Splenocytes were cultured in TCM from NT or STm-infected tumours and intracellular asparagine synthetase (ASNS) expression was analysed in CD4+ and CD8 + T cells 24–48 h after stimulation with a-CD3/a-CD28 antibodies (1 and 5 mg/mL, respectively). Each point represents an independent spleen donor, $n = 3$. Bars depict means ± SEM or median with interquartile range. Statistical significance was tested by two-way ANOVA with Sidak's post-test (E, F, I) or two-way ANOVA with Tukey's post-test (H). Source data are available online for this figure.

then adding $1 \times 10^5$ bacteria (which is approximately equal to the CFU we recover from live infection) to the tumour media for the duration of the culture.

## Ex vivo tumour culture

To culture primary tumours, tissue was excised from the CAC model as described and then cut into 1–2 mm pieces using sterile scissors. Equivalent amounts of tumour pieces were then resuspended in a 20–50 μL Matrigel dome and cultured in either tumour organoid media or TCM. Antibodies (1 μg/mL α-CD3 and 5 μg/mL α-CD28) were added to the appropriate wells in order to stimulate TILs, since these can penetrate the gel dome and access the tumour tissue (Voabil et al, 2021). For cytokine polyfunctionality assays, 10 μg/mL brefeldin A (Merck) was added during the 6 h stimulation; for the *Nr4a3*-Timer experiment, antibodies alone were used for 16 h. After stimulation, Matrigel was dissolved using 500 μL Cell Recovery Solution (Corning) for 30 min at 4 °C, followed by washing and then processing tumours as described for tumour digestion and TIL analysis.

## Splenocyte isolation

Splenocytes were isolated by mechanical disruption of the spleen, filtering through a 70-μm strainer, and then red blood cell lysis by ACK Lysis Buffer (Invitrogen) for 5 min on ice. Cells were then re-filtered through a 70-μm strainer and kept on ice until further use.

## Tumour organoid—splenocyte/T cell co-culture

To assess T cell activation in the TME, splenocytes were directly cultured with tumour organoids after the 2 h infection window with STm. Cells were seeded in the tumour wells at $1–2 \times 10^6$ per well for the time indicated in the figure legends, and the appropriate wells were stimulated with 1 μg/mL α-CD3 and 5 μg/mL α-CD28. Alternatively, cells were stimulated in 96-well round-bottom plates in the TCM from the end of a 48 h tumour infection. This TCM was frozen at −20 °C until use. Data points may indicate either individual mice or individual infections from pooled splenocytes, as specified.

To some conditions, the following were added at the indicated timepoints: c-Myc inhibitor (10058-F4; Selleck Biochem), amino acids (10 mM glutamine, asparagine or aspartic acid—all from Merck), or D-glucose (50 mM; Merck).

## Proliferation assay

To measure proliferation, splenocytes were stained with CellTrace Blue (Invitrogen) according to manufacturer instructions and were then cultured in 96-well round-bottom plates in 100 μL TCM for 3 days ± activating antibodies (1 μg/mL α-CD3 and and 5 μg/mL α-CD28). Cells were then acquired on the flow cytometer and dye-low cells were indicative of proliferating cells.

## Enzyme-linked immunosorbent assay

Splenocytes ($1 \times 10^6$) were cultured in 96-well round-bottom plates in 100 μL TCM. Some conditions were activated using 1 μg/mL α-CD3 and 5 μg/mL α-CD28. Supernatants were collected after 24 h and extracellular IL-2 was measured using the Biolegend ELISA-Max Deluxe Kit (Biolegend), according to manufacturer instructions.

## General flow cytometry

Flow cytometry was performed in 96-well round-bottom plates. Cells were transferred to plates, and then washed in 100–200 μL FACS buffer (PBS containing 2% FCS, 2 mM EDTA (Merck) and centrifuged at 400×*g* for 5 min, after which the supernatant was discarded. Cells were then stained in the appropriate antibody master mix containing 1:1000 eFluor-780 Fixable Viability Dye (Invitrogen) and 1:500 Fc block (Prof. Anne Cooke, University of Cambridge). Cells were stained for 30 min at 4 °C while protected from light, then washed in FACS buffer and acquired on a BD Fortessa X20. UltraComp eBeads (Invitrogen) and unstained cells were used for compensation controls.

## Intracellular flow cytometry

For intracellular stains, cells were surface stained as described above, but then pelleted and resuspended in 100 μL fixative from the eBioscience Foxp3/Transcription Factor Staining Buffer Set (Invitrogen) for 30 min at 4 °C while protected from light. Fixative was then washed off and discarded, and next cells were washed in FACS buffer followed by the kit permeabilisation buffer (Invitrogen). Cells were then resuspended in a staining mixture made up of permeabilisation buffer and stained for 30–60 min at 4 °C in

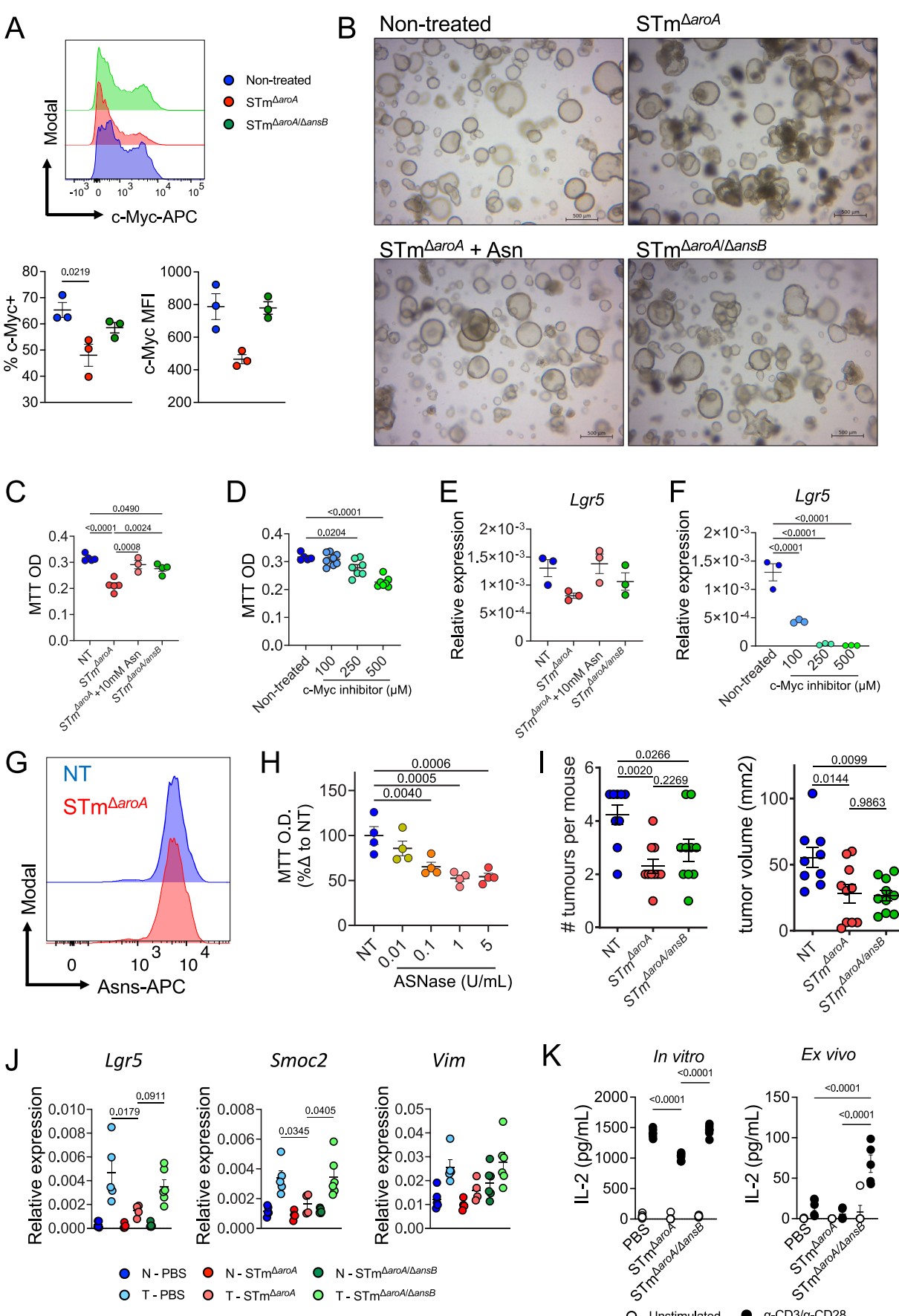

◄ **Figure 8. Deletion of bacterial asparaginase *ansB* result in loss of tumour c-Myc control but retains in vivo tumour suppression.**

(A) Tumour organoids were infected with STm$^{\Delta aroA}$ or STm$^{\Delta aroA/\Delta ansB}$ for 24–48 h and then stained for intracellular c-Myc. Data pooled from two independent experiments ($N = 2$ and $N = 1$, technical replicates). Each dot represents an individual well of tumour organoids. (B,C) Tumour organoids were infected with the indicated STm mutant with or without asparagine supplementation for 24 h. Organoid viability/metabolic capacity was assessed by MTT assay (C). (B) Representative images of organoids after 24 h treatment. Dark blebbing organoids are dying. Each dot represents an individual well (technical replicate), $n = 5$ NT& STm$^{\Delta aroA}$, $n = 4$ STm$^{\Delta aroA/\Delta ansB}$, $n = 3$ STm$^{\Delta aroA}$ + 10 mM Asn. Representative of two independent experiments using two different Apc$^{min/+}$-derived tumour organoid lines. (D) Tumour organoids were treated with a c-Myc inhibitor for 24 h. Organoid viability/metabolic capacity was assessed by MTT assay. Each dot represents an individual well (technical replicate). $n = 5$ (NT), 8 (100 μM), 7 (250 & 500 μM). Representative of two independent experiments using two different Apc$^{min/+}$-derived tumour organoid lines. (E, F) RNA was isolated from organoids in (C, D) and qPCR for *Lgr5* performed. $n = 3$, all groups. Each dot represents an individual well of tumour organoids (technical replicates). (G) NT or STm-infected tumour organoids (24 h) were stained for asparagine synthetase and analysed by flow cytometry. Representative plot of $N = 2$. (H) Tumour organoids were cultured with the indicated concentrations of *E.coli*-derived asparaginase for 24 h and growth was assessed by MTT assay. $n = 4$, each group. Representative of two independent experiments, each conducted on two independent Apc$^{min/+}$-derived tumour organoid lines. Each dot represents an individual well of tumour organoids (technical replicates). (I) CAC was induced in C57BL/6 mice using AOM/DSS followed by weekly oral gavage with $5 \times 10^9$ CFU of STm$^{\Delta aroA}$, STm$^{\Delta aroA/\Delta ansB}$ or PBS control for 6 weeks, and tumour volume were assessed. Each dot represents one mouse ($n = 10$ per STm group, 9 PBS group). (Data of NT v's STm$^{\Delta aroA}$ taken from Mackie et al, 2021, rep of .five experiments), ansB treatment group representative of two experiments. (J) RNA was isolated from the normal colon (N) or tumour (T) from experiment in (I) and qPCR analysis of indicated stem/mesenchymal transcripts performed. Each dot represents tumour from one mouse, data representative of >3 experiments for the PBS and STm$^{\Delta aroA}$ groups and 1 experiment for the STm$^{\Delta aroA/\Delta ansB}$ group. (Data of NT v's STm$^{\Delta aroA}$ taken from Mackie et al, 2021). $n = 5$ NT, $n = 4$ STm$^{\Delta aroA}$, n-6 STm$^{\Delta aroA/\Delta ansB}$. (K) IL-2 measured by ELISA from in vitro co-cultured splenocytes activated by a-CD3/28 in the presence of TCM from indicated STm strains ($n = 6$; pooled TCM tested on six independent spleen donors), or ex vivo re-stimulated TILs ($n = 5$; each dot representing TILs isolated from independent mice). Bars depict means ± SEM or median with interquartile range. Statistical significance was tested by one-way ANOVA with Tukey's or Dunnett's post-test (**C-F, H, I, K**), or Kruskal–Wallis with Dunn's post-test (**A, J**). Source data are available online for this figure.

complete darkness. Next, cells were washed in a permeabilisation buffer, and then in FACS buffer before data acquisition. However, antibodies requiring a secondary conjugation were instead stained for another 30 min in permeabilisation buffer as an extra step, before washing and acquisition.

Flow cytometry of c-Myc in tumour organoids was performed by first dissolving Matrigel using Cell Recovery Solution (Corning), and then digesting the organoids into single cells using 1 mL TryplE per condition at 37 °C for 5 min, followed by washing and proceeding as normal. All flow cytometry steps before fixation with tumour organoids were performed in the presence of ROCK inhibitor (Y-27632) used at 5 μM.

### PhosFlow

To stain phosphorylated epitopes, cells were stained according to surface stain procedures, but transferred to 5 mL polystyrene tubes and then washed twice in excess PBS. Cells were then resuspended in 1:1 PBS-diluted Fixation Buffer (Biolegend), comprising ~2% PFA, and incubated at 37 °C in a water bath with gentle agitation. Following mild fixation, cells were centrifuged (1000×*g* for 5 min), PFA was discarded, and then cells were washed twice with PBS containing 2% FCS. The cell pellet containing residual buffer was vortexed, and then 1 mL TruePhos Fixation Buffer (Biolegend) was added drop-by-drop while under constant vortexing. Samples were then stored at −20 °C in the dark for 1 h. Samples were next centrifuged, and the fixation buffer discarded, followed by two washes in PBS/FCS. Finally, cells were stained with phospho-antibodies in PBS/FCS for 30–45 min at 4 °C in complete darkness, followed by a wash and then acquisition on a flow cytometer.

### Protein translation assay

Protein translation was measured by activating splenocytes ($1 \times 10^6$) in 100 μL TCM from infected or non-infected tumours, for the designated times (with the longest stimulation being performed first). At the end of the assay, 5 μg/mL puromycin

dihydrochloride (Merck) was added to each well, and the plate was incubated for 15 min at 37 °C. Next, cells were stained for surface markers as described, followed by fixation and intracellular staining using an anti-puromycin antibody (1:200). Cells were then analysed by flow cytometry, and suppression of protein translation was calculated relative to the control.

### Calcium flux assay

Splenocytes ($1 \times 10^6$) were cultured with TCM from infected or non-infected tumours for 24 h. Next, cells were surface stained for flow cytometry in 5 mL polystyrene tubes as described, washed in PBS, and then stained with 2.5 μM Fura-Red and Flura-4 mixture (both Invitrogen) for 30 min at 37 °C. Cells were then washed twice in PBS containing 2% FCS to quench any extracellular dye, rested for 10 min, followed by one wash in pure PBS. Cells were then acquired on the flow cytometer for ~25 s before the addition of 1 μg/mL ionomycin (Merck), and calcium flux was measured for ~1.5 min.

### Nuclear flow cytometry

Analysis of transcription factors within T cell nuclei was performed according to an established protocol (Gallagher et al, 2021, 2018). Bulk T cells were first negatively isolated using either MojoSort (Biolegend) or MACS separation (Miltenyi Biotech) to high purity (>97%), according to manufacturer instructions. Next, T cells were stimulated in TCM from non-infected or STm-infected tumours for 24 h. One hour before the end of the culture, 1 μg/mL α-CD3 and 5 μg/mL α-CD28 were added to stimulated conditions. After stimulation, cells were centrifuged, media was discarded, and the plate was put on ice. Sucrose buffer (200 μL/well) containing 10 mM HEPES, 8 mM MgCl$_2$ (Merck), 320 mM sucrose, 1 x EDTA-free protease inhibitor (Roche) and 0.1% Triton X-100 (Merck) was added to each well, and cells were incubated for 15 min. The plate was then centrifuged at 2000×*g* for 5 min while chilled, and the supernatant was discarded. Next, cells were washed twice in the sucrose buffer (lacking Triton X-100), followed by fixation in 4% PFA diluted in sucrose buffer (again lacking

Triton X-100) for 30 min on ice, protected from light. Cells were washed, and the supernatant discarded. Cells were then washed again in FACS buffer containing 8 mM $MgCl_2$, followed by a wash in permeabilisation buffer (FACS buffer with 0.3% Triton X-100 and 8 mM $MgCl_2$). Cells were stained in antibodies diluted in permeabilisation buffer for 1 h, before washing with magnesium-supplemented FACS buffer and acquisition on the flow cytometer. Cells were stained for β-tubulin to check the purity of nuclei, which also served as a gate to exclude any cytoskeletal debris.

## RNA extraction and RNAseq

Splenocytes were activated in TCM from infected or non-infected tumours as described, and then at 4 or 16 h, CD4 T cells were fluorescently sorted to high purity on a FACSAria. RNA was extracted from lysates using the Arcturus Picopure RNA kit (ThermoFisher) according to the manufacturer's instructions. About 5 ng of RNA was used for the generation of sequencing libraries using the Quantseq 3' mRNA-seq Library Preparation kit FWD (Lexogen). Unique Molecular Identifiers (UMIs) were used for the evaluation of input and PCR duplicates and to eliminate amplification bias. Libraries were normalised and pooled at a concentration of 4 nM for sequencing. Libraries were sequenced using the NextSeq 500 using a Mid 150v2.5 flow cell. Cluster generation and sequencing was then performed, and FASTQ files were generated. FASTQ files were then downloaded from the Illumina base space and uploaded to the BlueBee cloud for further analysis (Lexogen). FASTQ files were merged from the four lanes to generate final FASTQ files, which were loaded into the BlueBee QuantSeq FWD pipeline and aligned to the GRCm38 (mm10) genome. HTSeq-count v0.6.0 was used to generate read counts for mRNA species and mapping statistics. Raw read counts in the .txt format were used for further analysis using DESeq2 (Love et al, 2014) in R version 4.0. A DESeq dataset was created from a matrix of raw read count data. Data were filtered to remove genes with fewer than ten reads across all samples. Log2 fold change estimates were generated using the DESeq algorithm, and shrinkage using the ashr algorithm (Stephens, 2017) to estimate log2 fold changes (lfc). Differentially expressed genes (DEGs) were selected based on an adjusted $p$ value of <0.05, and then gene lists filtered to identify genes with an estimated lfc greater or less than 1.5. Normalised read counts were transformed using the regularised log (rlog) transformation. Heatmap analysis was performed on the rlog transformed data using the R package gplots. For KEGG pathway analysis clusterProfiler (Yu et al, 2012), DOSE (Yu et al, 2015), and biomaRt (Durinck et al, 2009) packages were used.

## ECAR and OCR analysis

To measure extracellular acidification (ECAR) and oxygen consumption rate (OCR), splenocytes ($2 \times 10^6$/well) were activated in technical triplicate from each mouse for 48 h in TCM from non-treated or STm-infected tumour organoids. Cells were then washed and resuspended in 'Seahorse XF' serum-free RPMI (Agilent), seeded onto pre-coated (poly-D-lysine; Invitrogen) Seahorse plates and analysed on a Seahorse XFe96 metabolic extracellular flux analyser as previously described (Gudgeon et al, 2022). Glucose (10 mM Sigma) was first added, and then Oligomycin (1 μM; Merck), BAM-15 (3 μM; Merck), rotenone (2 μM; Merck) and antimycin A (2 μM; Merck) were injected to disrupt various elements of the metabolic pathways. Basal OCR was calculated as the mean of the initial three measurements minus the mean of the

three measurements after rotenone/antimycin A injection. ATP-coupled OCR was calculated as the basal OCR minus the mean of the three measurements after oligomycin injection. Maximal OCR was calculated as the mean of the three measurements after the BAM-15 injection minus the mean of the three measurements after rotenone/antimycin A injection. Spare respiratory capacity was calculated as maximal OCR minus basal OCR. Glycolysis was calculated as the mean of the three measurements after glucose injection minus the mean of the initial three measurements. Glycolytic capacity was calculated as the mean of the three measurements after oligomycin injection minus the mean of the initial three measurements. The glycolytic reserve was calculated as the glycolytic capacity minus glycolysis.

## Tumour organoid qPCR

Tumour organoids were infected as described above. qPCR was performed as previously described (Mackie et al, 2021). Briefly, the culture medium was aspirated from treated organoids and Matrigel domes washed with PBS. RNA lysis buffer was added directly to the Matrigel domes, lysing Matrigel and organoids in one step. RNA was then isolated using a commercial kit (Qiagen RNeasy Mini Kit, Qiagen) according to manufacturer instructions. m-MLV and oligoDTs (Merck) were used to generate cDNA, and qPCR was performed using a Roche LightCycler 480 with SYBR green system (Takara). Below are the primers used for this study:

*Lgr5*: 5'-CGGAAAGTGGAATCCTTGCA, 3'-CACATCGATCTGGACATGCTGT

*Vim*: 5'-CGGAAAGTGGAATCCTTGCA, 3'-CACATCGATCTGGACATGCTGT

*Smoc2*: 5'-CCCTCAGAAGCCACTCTGTG, 3'-ACTTGCTGGAACTCCTTCCG

*Gapdh*: 5'-TGTGTCCGTCGTGGATCTGA, 3'-TTGCTGTTGAAGTCGCAGGAG

## MTT assay

Tumour organoids were infected as previously described and dissociated using Cell Recovery Solution and TryplE, and then seeded into 96-well flat bottom plates at $4 \times 10^4$/well in 10 μL Matrigel domes. 150 μL tumour organoid media was placed on top of each dome. After 24 h, 50 μL media was removed and replaced with 50 μL MTT reagent (Abcam), which was incubated for 1–3 h at 37 °C and 5% $CO_2$. Upon the appearance of purple crystals, the assay was halted by the addition of 4 mM HCl, 0.1% NP40 in isopropanol. Optical density was then measured by plate-reader at $OD_{560}$. Experiments were performed in triplicate and averaged.

## Software and statistical analysis and sample size

Data were analysed primarily using FlowJo version 9, GraphPad Prism version 9 and Microsoft Excel. RNAseq analysis was performed using software described in the relevant section. Graphical figures were made using BioRender (BioRender.com) and Microsoft Powerpoint. Statistical analyses are specified in the relevant figure legends, with appropriate post-tests to control for multiple testing. Data from individual mice in some figures were excluded if recovered tumour cell populations were of insufficient size for detection. Statistical significance was determined as being $p \leq 0.05$.

### The paper explained

#### Problem

Treatment options for colorectal cancer (CRC) are still relatively limited, and CRC is largely refractory to immune checkpoint blockade therapies due to poor immune infiltration. Use of bacterial cancer therapies is one approach to induce anti-tumour immune responses, and there is clinical interest in the potential of combining bacterial therapies with immune checkpoint inhibitors. However, the way the adaptive immune response—T cells in particular—functions in *Salmonella* cancer therapies remains largely unknown, and previous studies in mice suggest that T cells do not play any role in the efficacy of *Salmonella*-mediated cancer therapy. Here, we aimed at understanding how T cells function in *Salmonella* cancer therapy using mouse models of CRC.

#### Results

Tumours treated with an attenuated *Salmonella* typhimurium strain ($STm^{\Delta aroA}$) exhibited impaired activation of both CD4+ and CD8+ T cells. T cells activated in the presence of $STm$-infected tumours showed reduced proliferation and production of cytokines in ex vivo tumours and co-culture models with tumour organoids. T cell receptor signalling was initially intact, but full activation could not be sustained due to an inability to induce metabolic reprogramming. This was underpinned by the failure to sustain c-Myc protein levels, and c-Myc inhibition was driven by *Salmonella*-induced asparagine depletion. Removing *Salmonella* asparaginase activity restored asparagine levels and T cell activity. This, however, came at the expense of suppressing tumour cell c-Myc activities and effects on the tumour stem cell compartment. Despite this, the use of $STm^{\Delta aroA}$ also lacking asparaginase reduced tumour burden in a model of colitis-associated CRC.

#### Impact

This work has improved our understanding of T cell functionality during $STm$ cancer therapy and is further applicable to other bacterial strains. Our findings will contribute to the rational engineering of $STm$-cancer therapies and co-therapy indications.

No blinding was performed. Sample sizes for in vivo treatment with STm were estimated based on previous experiments, with tumour burden as an outcome, as a guide. Based on an effect size of 50% reduction in tumour burden (NT group average of six tumours with a 1.5% STDev), an unpaired, two-sided T-test with 90% power with 0.05 alpha-error probability a group size of 7 was required. Sample sizes were estimated for ex vivo TIL analysis based on the above, but also driven by a number of samples that can be processed and analysed in a timeframe that will not lose fluorescent timer. In vitro sample sizes, in general, were $n = 3$ biological (or technical where indicated) replicates per experiment and reproduced in independent experiments (indicated for each experiment).

## Data availability

RNA-Seq data: Gene expression omnibus: https://www.ncbi.nlm.nih.gov/geo/query/acc.cgi?acc=GSE218978. Flow cytometry source .fcs files: https://zenodo.org/records/13740695.

The source data of this paper are collected in the following database record: biostudies:S-SCDT-10_1038-S44321-024-00159-2.

## Peer review information

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

## Acknowledgements

We thank Professor Adam Cunningham and Professor David Withers (University of Birmingham, UK) for providing the SL3261 auxotrophic aroA *Salmonella* strain and attenuated Listeria monocytogenes strain, respectively. Ardeypharm GmbH for the provision of *E. coli* Nissle 1917 strain. University of Birmingham Biological Services Unit. Dr David Sumpton and the Metabolomics Core Facility at CRUK Scotland Institute for metabolomics analysis. Dr Rebecca Drummond (University of Birmingham, UK) and Dr Wei Li-Yu (University of Edinburgh, UK) for constructive feedback on the work. We also thank Professor Leslie Berg (University of Colorado, US) for providing the nuclei isolation protocol. Professor Anne Cooke (University of Cambridge, UK) generously provided functional antibodies. Work was funded by a CRUK Career establishment award (C61638/A27112, to K.M.M.), MRC CDA award (MR/V009052/1, to D.B.). BBSRC David Phillips Fellowship (BB/J013951, to M.O.) funded work to generate Nr4a3-Timer-of-cell-kinetics-and-activity mice.

## Author contributions

**Alastair Copland**: Conceptualisation; Resources; Data curation; Formal analysis; Supervision; Validation; Investigation; Visualisation; Methodology; Writing—original draft; Project administration; Writing—review and editing. **Gillian M Mackie**: Resources; Investigation; Methodology. **Lisa Scarfe**: Investigation. **Elizabeth Jinks**: Investigation. **David A J Lecky**: Resources. **Nancy Gudgeon**: Resources; Methodology. **Riahne McQuade**: Investigation. **Wilma H M Hoevenaar**: Investigation. **Masahiro Ono**: Resources; Methodology. **Manja Barthel**: Resources. **Wolf-Dietrich Hardt**: Resources. **Hiroshi Ohno**: Resources. **Sarah Dimeloe**: Investigation; Visualisation; Methodology. **David Bending**: Resources; Data curation; Formal analysis; Visualisation; Methodology; Writing—review and editing. **Kendle M Maslowski**: Conceptualisation; Resources; Data curation; Formal analysis; Supervision; Funding acquisition; Validation; Investigation; Visualization; Methodology; Writing—original draft; Project administration; Writing—review and editing.

Source data underlying figure panels in this paper may have individual authorship assigned. Where available, figure panel/source data authorship is listed in the following database record: biostudies:S-SCDT-10_1038-S44321-024-00159-2.

## Disclosure and competing interests statement

The authors declare no competing interests.

# Expanded View Figures

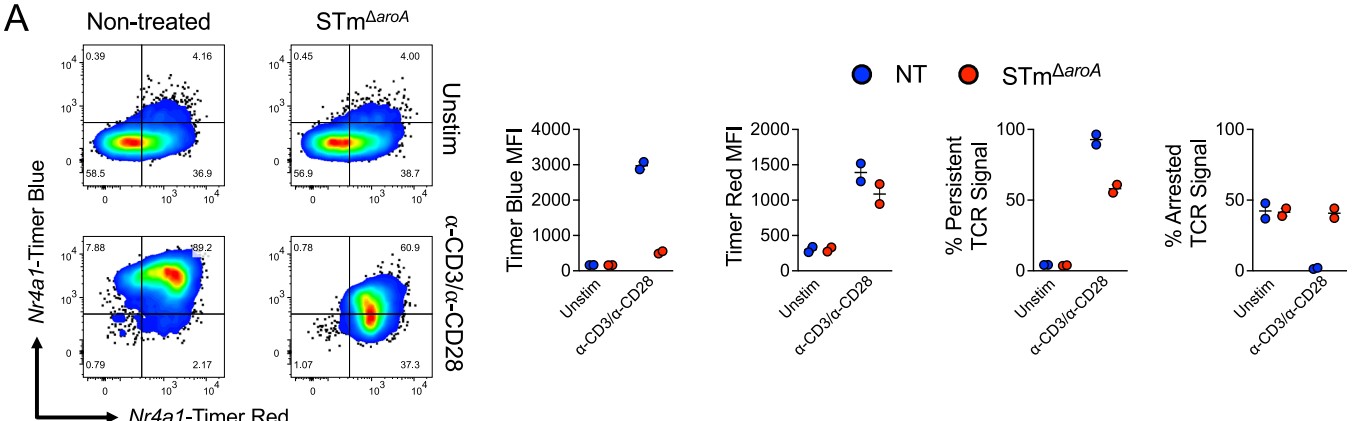

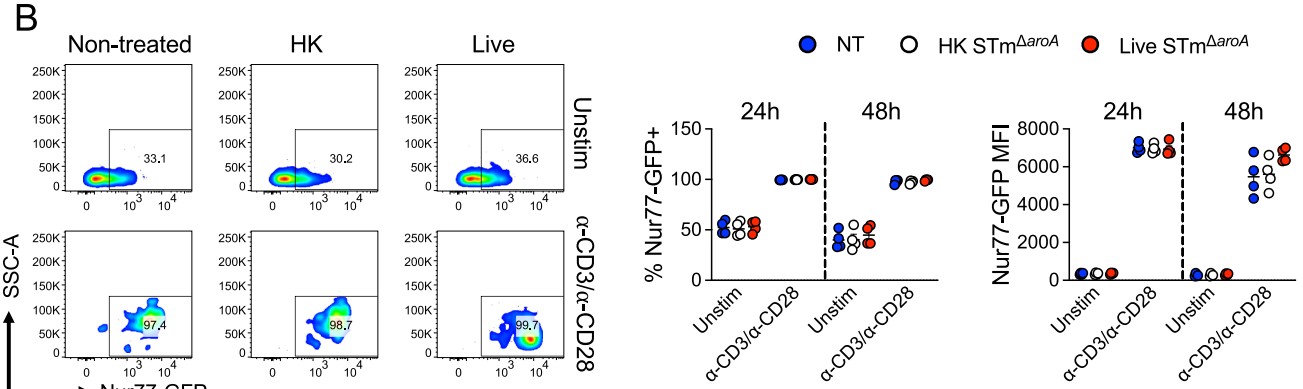

**Figure EV1. Differential sensitivity of Timer and GFP-based TCR reporters in detecting arrested T cell activation.**

TCM from NT or STm-infected tumour organoids was used to culture splenocytes from *Nr4a1*-Tempo (A) or *Nur77*-GFP (B) T cell reporter mice for 24 h (Tempo mice) or up to 48 h (Nur77 mice) after stimulation with α-CD3/α-CD28 (1 and 5 mg/mL, respectively). Data depict *n* = 2 (**A**) or *n* = 3 (**B**) mice.

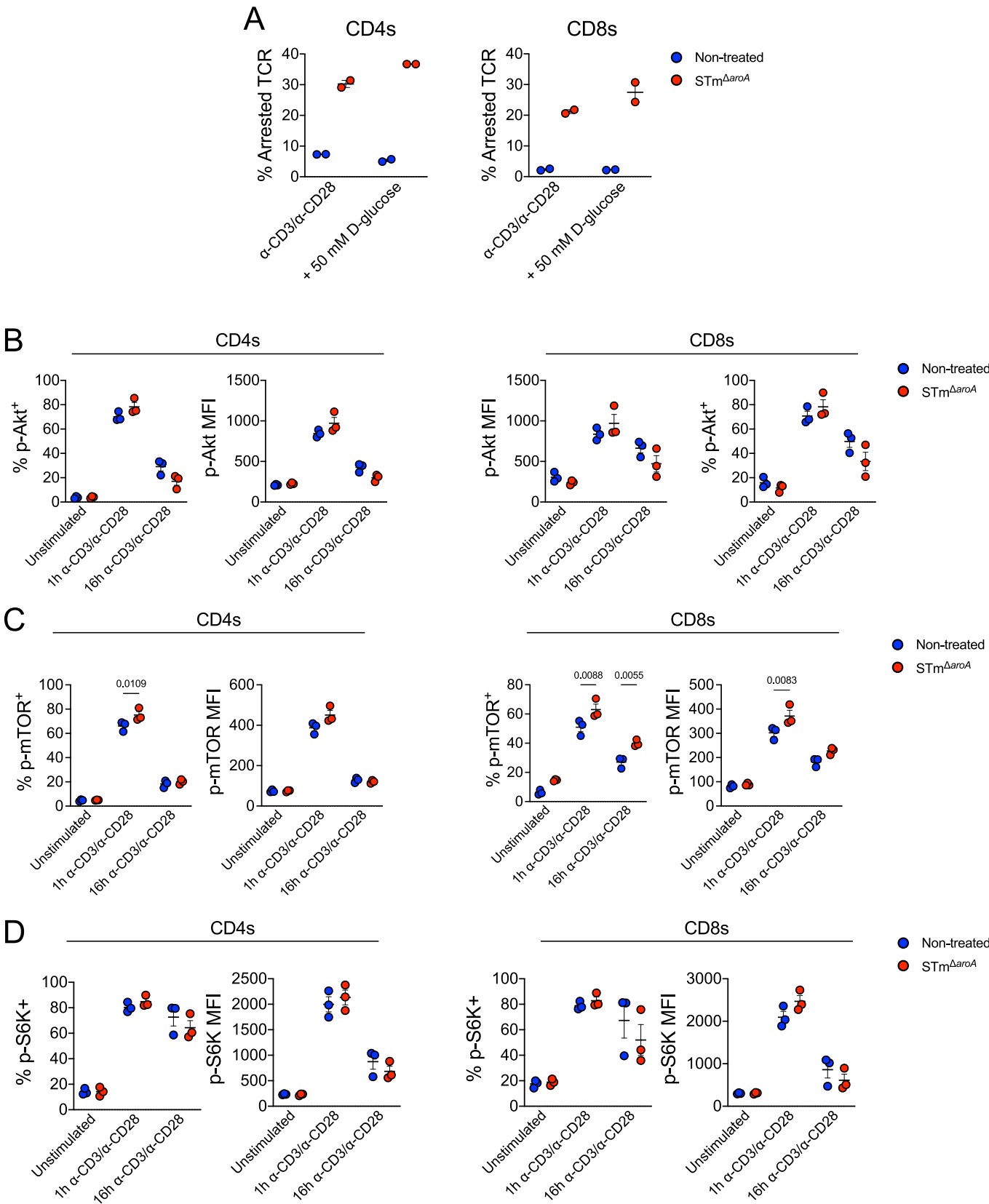

◀ **Figure EV2. T cells activated in the presence of STm-infected tumours are unable to increase glucose uptake and show normal TCR signalling cascades.**

(A) Splenocytes were co-cultured with infected or non-treated tumours for 24 h in the presence of α-CD3/α-CD28 antibodies (1 and 5 µg/mL, respectively). During the final 6 h, a high dose of glucose (50 mM) was spiked into the culture in an attempt to reduce arrested TCR activation, as measured by *Nr4a3*-Timer Arrested TCR signal, i.e. *Nr4a3*-Timer Blue^neg Timer Red^pos. Cells were then tested for activation by flow cytometry. Data show two independent infections; each data point represents splenocytes tested with an independent tumour infection. (B–D) Quantification of PhosFlow signalling pathways as shown by representative plots in Fig. 6E. Cells were stimulated in NT and STm TCM for 1 or 16 h and processed for PhosFlow as previously described, showing p-Akt-S473 (B), p-mTOR-S2448 (C) and p70-S6K-T421/S424 (D). Bars depict means ± SEM. Statistical significance was tested by two-way ANOVA with Sidak's post-test (non-treated vs STm). Data were derived from $n = 3$ mice testing pooled TCM from two tumour infections.

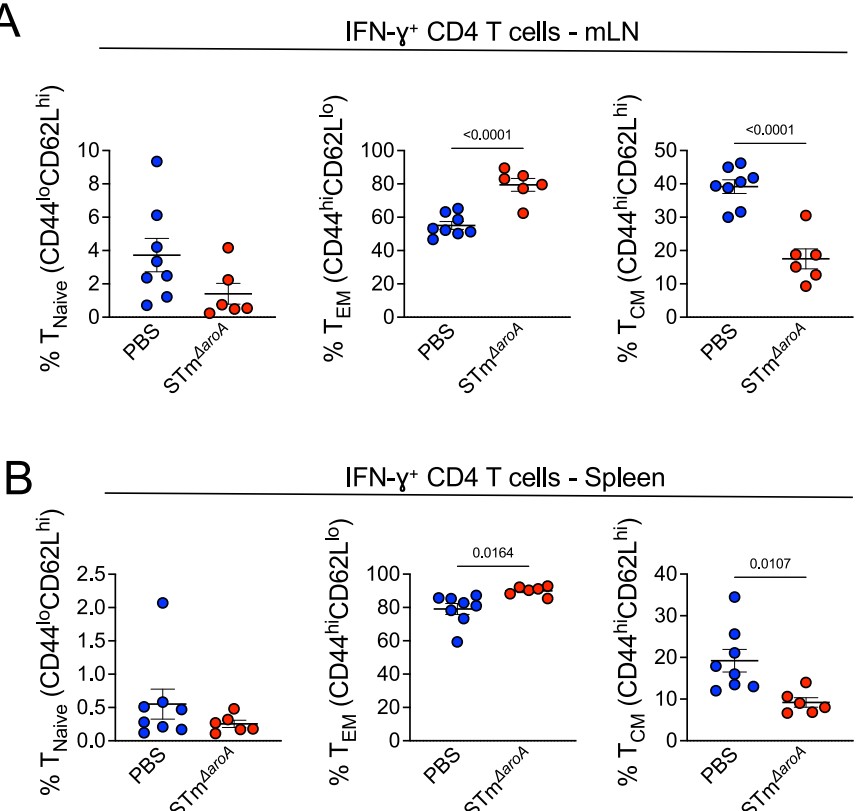

**Figure EV3.   Activated lymphatic T cells from mice infected with STm show a T_EM-bias.**

Tumours were induced in mice using the CAC model, followed by two rounds of oral STm treatment as previously outlined for immunogenicity experiments. One week after the final dose, mice were culled and mLN and spleens were extracted, followed by preparation of single-cell suspensions and flow cytometry staining for T cell memory subsets within the IFN-γ⁺ CD4 (**A**) and CD8 T cell (**B**) populations. Cells were phenotyped based on the expression of CD44 and CD62L. Bars depict means ± SEM. Each data point represents one mouse, $n = 8$ NT, $n = 6$ $STm^{\Delta aroA}$. Statistical significance was tested by unpaired two-tail $t$-test.

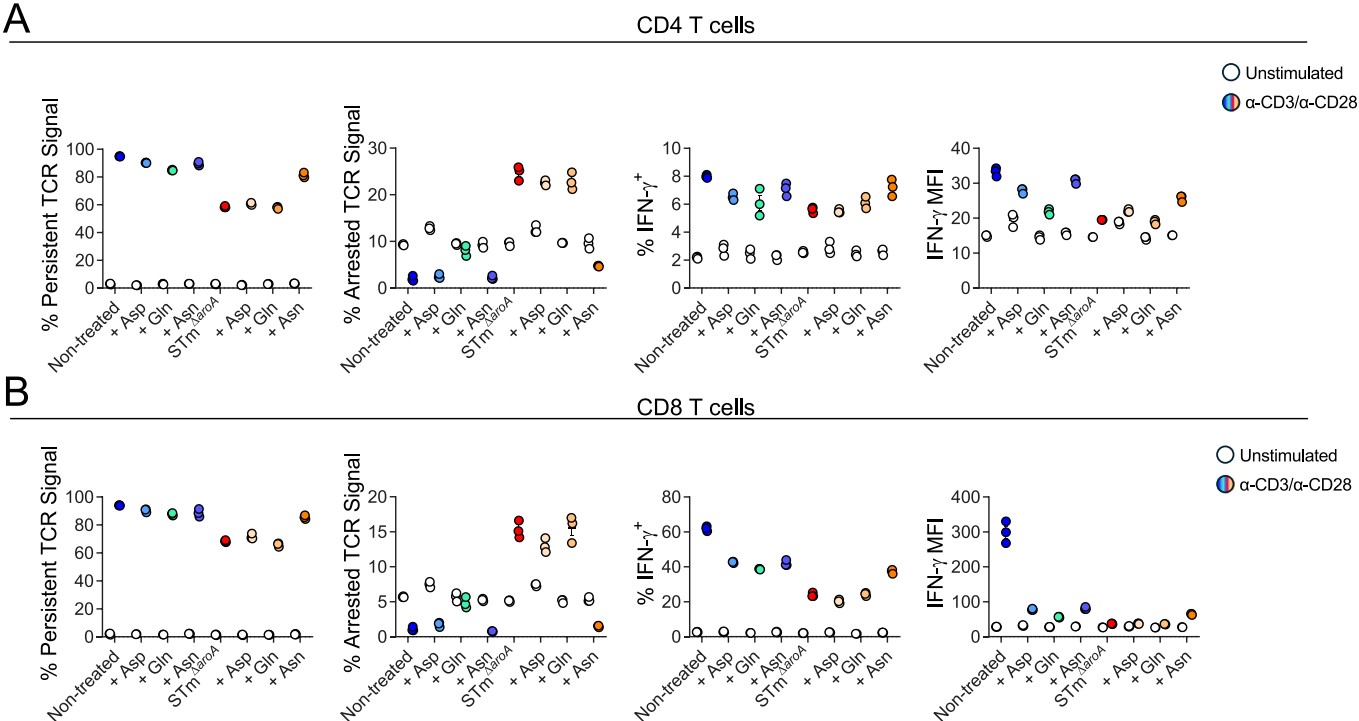

**Figure EV4. Asparagine supplementation is able to restore T cell activation when cultured with TCM from infected tumours.**

Splenocytes were cultured in TCM from either non-infected or infected-tumour organoids and activated for 24 h with α-CD3/α-CD28 antibodies (1 and 5 μg/mL, respectively). To some cultures, Asp/Gln/Asn were added (10 mM) at the beginning of the culture, or else a vehicle control ($H_2O$) was used. Various metrics of either *Nr4a3*-Timer, indicative of TCR signalling, or IFN-g expression were quantified by flow cytometry for CD4 (**A**) and CD8 (**B**) T cells. Data are from $n = 3$ mice, using pooled TCM from two tumour infections. Bars depict means ± SEM.

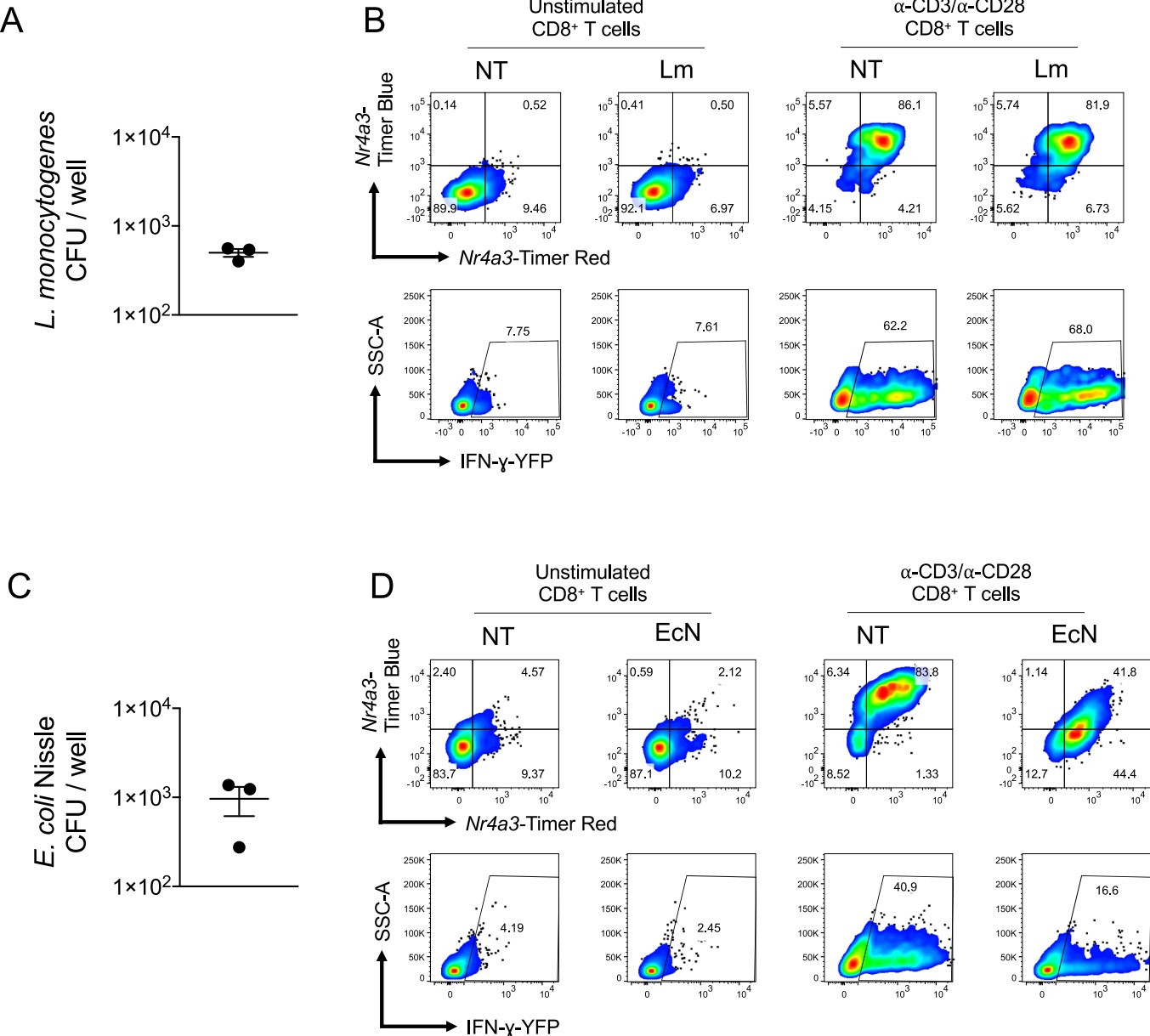

**Figure EV5. Oncolytic bacteria possessing the high activity ASNase gene also suppress T cell Activation.**

Tumour organoids were infected with either *L. monocytogenes* (Lm) or *E. coli* Nissle (EcN) and the TCM was used to stimulate splenocytes. (**A**) CFU assessment of Lm 24 hr after infection. Each dot represents an individual well of organoids, $n = 3$. (**B**) *Nr4a3*-Timer and IFN-γ expression of unstimulated or a-CD3/a-CD28 stimulated (1 and 5 mg/mL, respectively) CD8 + T cells from activated splenocytes cultured in Lm TCM. Data representative flow plots of $n = 3$ independent spleen donors. (**C**) CFU assessment of E.coli Nissle after 24 h infection. Each dot represents an individual well of organoids, $n = 3$. (**D**) *Nr4a3*-Timer and IFN-γ expression of unstimulated or a-CD3/a-CD28 stimulated (1 and 5 mg/mL, respectively) CD8 + T cells from activated splenocytes cultured in EcN TCM. Data representative flow plots of $n = 3$ independent spleen donors. Error bars depict SEM.

