## [Peer Review File · EMBO Molecular Medicine]

Salmonella cancer therapy metabolically disrupts tumours at the collateral cost of T cell immunity

Alastair Copland, Gillian Mackie, Lisa Scarfe, Elizabeth Jinks, David Lecky, Nancy Gudgeon, Riahne McQuade, Wilma Hoevenaar, Masahiro Ono, Manja Barthel, Wolf-Dietrich Hardt, Hiroshi Ohno, Sarah Dimeloe, David Bending, and Kendle Maslowski

Corresponding authors: Kendle Maslowski (kendle.maslowski@glasgow.ac.uk) , Alastair Copland (a.copland@bham.ac.uk)

Review Timeline:

Submission Date:	17th May 24
Editorial Decision:	5th Jun 24
Revision Received:	10th Sep 24
Editorial Decision:	17th Sep 24
Revision Received:	4th Oct 24
Accepted:	11th Oct 24

Editor: Lise Roth

Transaction Report:

Please note that the manuscript was previously reviewed at another journal and the reports were taken into account in the decision making process at EMBO Molecular Medicine. Since the original reviews are not subject to EMBO's transparent review process policy, the reports and author response cannot be published.

5th Jun 2024

Dear Dr. Maslowski,

Thank you for the submission of your manuscript to EMBO Molecular Medicine, following peer-review and rejection at a different journal.

Briefly, two of the initial referees agreed that the study was interesting, well-performed and included high-quality models. They nevertheless raised concerns on novelty and requested complementary experiments and clarifications. The third referee was more critical, found the experiments poorly powered and the interpretation of the results not convincing.

We have sent your revised manuscript to a single reviewer for evaluation, together with the existing referees' reports, and your point-by-point rebuttal letter. We have now received the enclosed report from this reviewer, who is positive and supportive of publication pending minor revisions.

We will therefore welcome the submission of your revised manuscript according to this referee's recommendations.

EMBO Molecular Medicine encourages a single round of revision only and therefore, acceptance or rejection of the manuscript will depend on the completeness of your responses included in the next, final version of the manuscript. For this reason, and to save you from any frustrations in the end, I would strongly advise against returning an incomplete revision.

We require:

4) A .docx formatted letter INCLUDING the reviewers' reports and your detailed point-by-point responses to their comments. As part of the EMBO Press transparent editorial process, the point-by-point response is part of the Review Process File (RPF), which will be published alongside your paper.

5) A complete author checklist, which you can download from our author guidelines (<https://www.embopress.org/page/journal/17574684/authorguide#submissionofrevisions>). Please insert information in the checklist that is also reflected in the manuscript. The completed author checklist will also be part of the RPF.

6) Please note that all corresponding authors are required to supply an ORCID ID for their name upon submission of a revised manuscript.

7) It is mandatory to include a 'Data Availability' section after the Materials and Methods. Before submitting your revision, primary datasets produced in this study need to be deposited in an appropriate public database, and the accession numbers and database listed under 'Data Availability'. Please remember to provide a reviewer password if the datasets are not yet public (see <https://www.embopress.org/page/journal/17574684/authorguide#dataavailability>).

In case you have no data that requires deposition in a public database, please state so in this section ("This study includes no data deposited in external repositories.").

Note that the Data Availability Section is restricted to new primary data that are part of this study.

8) For data quantification: please specify the name of the statistical test used to generate error bars and P values, the number (n) of independent experiments (specify technical or biological replicates) underlying each data point and the test used to calculate p-values in each figure legend. The figure legends should contain a basic description of n, P and the test applied. Graphs must include a description of the bars and the error bars (s.d., s.e.m.). Please provide exact p values.

9) Our journal encourages inclusion of *data citations in the reference list* to directly cite datasets that were re-used and obtained from public databases. Data citations in the article text are distinct from normal bibliographical citations and should directly link to the database records from which the data can be accessed. In the main text, data citations are formatted as follows: "Data ref: Smith et al, 2001" or "Data ref: NCBI Sequence Read Archive PRJNA342805, 2017". In the Reference list,

data citations must be labeled with "[DATASET]". A data reference must provide the database name, accession number/identifiers and a resolvable link to the landing page from which the data can be accessed at the end of the reference. Further instructions are available at .

13) Author contributions: CRediT has replaced the traditional author contributions section because it offers a systematic machine readable author contributions format that allows for more effective research assessment. Please remove the Authors Contributions from the manuscript and use the free text boxes beneath each contributing author's name in our system to add specific details on the author's contribution. More information is available in our guide to authors.

16) As part of the EMBO Publications transparent editorial process initiative (see our Editorial at <http://embomolmed.embopress.org/content/2/9/329>), EMBO Molecular Medicine will publish online a Review Process File (RPF) to accompany accepted manuscripts.

In the event of acceptance, this file will be published in conjunction with your paper and will include the anonymous referee reports, your point-by-point response and all pertinent correspondence relating to the manuscript. Let us know whether you agree with the publication of the RPF and as here, if you want to remove or not any figures from it prior to publication. Please note that the Authors checklist will be published at the end of the RPF.

I look forward to receiving your revised manuscript.

Yours sincerely,

Lise Roth

***** Reviewer's comments *****

Referee #1 (Comments on Novelty/Model System for Author):

I found the work to be of quality and elegant, with a number of innovative models, particularly for the detailed monitoring of lymphocyte activation. It highlights the limitations of bacterial-based therapeutic approaches.

As one of the reviewers pointed out, the restoration of T lymphocyte function without any improvement in tumour growth inhibition is a little disappointing, but I feel that this negative result is important and opens up prospects for improving this approach.

It is a pity that this restoration of the T lymphocyte response by depleting asparaginase cannot be extended to the study of the specific anti-tumour T lymphocyte response and the associated memory, but the authors have clearly explained why their models could not address this criticism.

I therefore propose just a minor revision to clarify some points without additional experience as the previous 3 reviewers have already asked the important questions. I do not agree with the severity of reviewer C in particular on i) the originality of the manuscript compared to a previous work by the authors (Mackie et al 2021) which did not report the identification of asparaginase as an escape mechanism ii) the Thommen technique is very relevant for preserving tumour architecture iii) I am not disturbed by the transparent re-analysis of data if the results obtained are original.

Referee #1 (Remarks for Author):

It's a an elegant, high-quality piece of work using innovative tools.

The previous review was fairly extensive and I won't go back over the issues raised.

My few suggestions for improving the manuscript:

- Could the authors explain why PD-1, which is not only a marker of exhaustion but also of activation and regulated by NFAT, is not increased after bacterial infection? Is its expression increased after STm Δ aroA/ Δ ansB infection?
- In terms of translation to the human clinics, could the authors' results on the role of cMyc and asparaginase in bacterial infection enable us to consider a more specific clinical implications of this therapeutic approach (tumours less dependent on cMyc for their proliferation, etc.)?
- Give abbreviation of HK

Reply to Reviewers – EMBO Molecular Medicine – EMM-2024-19866

Copland et al, 2024, **Salmonella Cancer Therapy Metabolically Disrupts Tumours at the Collateral Cost of T cell Immunity**

***** Reviewer's comments *****

Referee #1 (Comments on Novelty/Model System for Author):

I found the work to be of quality and elegant, with a number of innovative models, particularly for the detailed monitoring of lymphocyte activation. It highlights the limitations of bacterial-based therapeutic approaches.

As one of the reviewers pointed out, the restoration of T lymphocyte function without any improvement in tumour growth inhibition is a little disappointing, but I feel that this negative result is important and opens up prospects for improving this approach.

It is a pity that this restoration of the T lymphocyte response by depleting asparaginase cannot be extended to the study of the specific anti-tumour T lymphocyte response and the associated memory, but the authors have clearly explained why their models could not address this criticism.

I therefore propose just a minor revision to clarify some points without additional experience as the previous 3 reviewers have already asked the important questions. I do not agree with the severity of reviewer C in particular on i) the originality of the manuscript compared to a previous work by the authors (Mackie et al 2021) which did not report the identification of asparaginase as an escape mechanism ii) the Thommen technique is very relevant for preserving tumour architecture iii) I am not disturbed by the transparent re-analysis of data if the results obtained are original.

We thank the reviewer for considerate assessment of the manuscript.

Referee #1 (Remarks for Author):

It's a an elegant, high-quality piece of work using innovative tools.

The previous review was fairly extensive and I won't go back over the issues raised.

My few suggestions for improving the manuscript:

- Could the authors explain why PD-1, which is not only a marker of exhaustion but also of activation and regulated by NFAT, is not increased after bacterial infection? Is its expression increased after STmΔaroA/ΔansB infection?

PD-1 is upregulated after sustained TCR activation. We think we don't see increased PD-1 after STm infection because the T cells are not activated long enough. We have not yet checked PD-1 expression following infection with STmΔaroA/ΔansB, but given we see sustained TCR signalling via the Timer protein analysis, and increased c-Myc and IL-2, we would assume an increase in PD-1 also. This will be part of ongoing studies, assessing the combination of STmΔaroA/ΔansB with immune checkpoint blockade inhibition.

We have added the following text (highlighted) to page 7:

Crucially, however, IFN- γ ⁺ TILs in *Salmonella*-treated mice exhibited a reduction in expression of PD-1 when compared to the PBS control group, in keeping with the TCR reporter showing

less evidence of TCR engagement (**Figure 1E**), as TCR engagement is necessary for upregulation of PD-1 (Elliot et al., 2021).

- *In terms of translation to the human clinics, could the authors' results on the role of cMyc and asparaginase in bacterial infection enable us to consider a more specific clinical implications of this therapeutic approach (tumours less dependent on cMyc for their proliferation, etc.)?*

This is a very pertinent point, and something we need to consider for future work. In the case of using an asparagine-deficient Salmonella, then yes, we predict a tumour less dependent may show more efficacy. In future studies we should consider timing of Salmonella dosing. Here, we continually dose weekly, so there will always be a presence of STm within tumours, affecting asparagine levels. In a previous study we had seen efficacy of treatment even with one dose We have added the following text to the discussion (page 25, highlighted):

Another consideration is whether certain tumours might respond better to bacterial therapy with an asparaginase-deficient STm, for example tumours with less reliance on c-Myc.

- *Give abbreviation of HK*

Now corrected in manuscript (page 9). Heat-killed (HK) Salmonella.

17th Sep 2024

Dear Dr. Maslowski,

Thank you for submitting your revised study. We have now received the report from the referee we consulted on your manuscript, who is satisfied with the revisions. I will therefore be able to accept your manuscript once the following editorial issues will be addressed:

1/ Manuscript text:

- Please remove the highlights in the text and only keep in track changes mode any new modification.
- Please note that all corresponding authors are required to supply an ORCID ID for their name upon submission of a revised manuscript. Currently, an email identifier is missing for A. Copland.
- The heading "Summary" should be updated to "Abstract".
- Please correct the order of the manuscript text as follows: Abstract / Keywords / Introduction / Results / Discussion / Methods / Acknowledgements / Disclosure and competing interests statement / The Paper Explained / References / Figure legends / Expanded View Figure legends
- Methods:
 - o Please provide a Reagents and Tools Table (listing key reagents, experimental models, software and relevant equipment and including their sources and relevant identifiers). Please download and fill our Reagents and Tools Table template (.docx), which you can find in our author guidelines: <https://www.embopress.org/page/journal/14693178/authorguide#structuredmethods>. When submitting your revised manuscript, please do not include the Reagents and Tools Table in the Methods section of the manuscript but upload it as a separate file choosing the file type "Reagent Table".
 - o Cells: please indicate whether the cells were tested for mycoplasma contamination.
 - o Antibodies: please provide dilutions/concentrations.
 - o Statistics: please provide statements on randomization. Please clarify "In vitro sample sizes estimated."
- Acknowledgements: The funding listed in this section should match the information entered in the submission system (the BBSRC David Phillips Fellowship (BB/J013951) is not entered in our system).
- Please provide a 'Disclosure statement and competing interests' section: We updated our journal's competing interests policy in January 2022 and request authors to consider both actual and perceived competing interests (<https://www.embopress.org/competing-interests>).
- References should list 10 authors before et al, and DOIs should be removed.

2/ Figures and Appendix:

- Figures need to be uploaded in EPS, TIF(F) or PDF format and in higher resolution. PPTX is not allowed. The correct nomenclature is "Figure EV1" etc.
- Please make sure that all figures/figure panels are referenced in the manuscript text (currently, a callout is missing for Fig. 5F)
- Please address the queries from our copy editors in the figure legends:
 1. Please note that the legends for figures EV 5b-c is not provided in the sequential manner (legend for figure EV 5c is provided before legend of figure EV 5b). This needs to be rectified.
 2. Please note that the legend for figure EV 2b-d is mislabeled as figure EV 2b-c in the manuscript. This needs to be rectified.
 3. Please note that the information on statistical test for figure 3f is mislabeled as 3h. This needs to be rectified.
 4. Please note that the exact p values are not provided in the legends of figures 1b-d; 3c-e, g; 5d; 6d, g; 7d-e; 8c-d, f, k; EV 3a.
 5. Please indicate the statistical test used for data analysis in the legends of figures 4c; 5f; 8k.
 6. Please note that for the figures 1b-e; 2b-c; 3c-g; 5a-f; 6a-d, f; 7d-f, h-i; 8a, c-d, f, h-k; EV 2c; EV 3a-b, p-values and statistical tests are indicated in the legends. However, comparison for the same, "****/**/*/*" has not been represented in the figures. Please rectify this in the figures or legends as applicable.
 7. Please note that n=2 in figures 6b, g; EV 1a; Ev 2a.
 8. Although 'n' is provided, please describe the nature of entity for 'n' in the legends of figures 8a, e-f, h, k.
 9. Please note that the error bars are not defined in the legends of figures EV 5a, c.

3/ Source Data:

- Figure 5C: identical values are provided for Non-treated + α -CD3/ α -CD28 and STm Δ aroA + α -CD3/ α -CD28; please clarify.
- Please double-check the images provided for Figure 8B.
- Figure 8J: identical values are provided for T-PBS and N - STm Δ aroA/ Δ ansB; please clarify.

4/ Checklist: Please check that you do not need to fill in the section 'DNA and RNA sequences'

5/ I included minor modifications in your Paper Explained. Please let me know if you agree or amend as you see fit:
Problem

Treatment options for colorectal cancer (CRC) are still relatively limited, and CRC is largely refractory to immune checkpoint

blockade therapies due to poor immune infiltration. Use of bacterial cancer therapies is one approach to inducing anti-tumour immune responses, and there is clinical interest in the potential of combining bacterial therapies with immune checkpoint inhibitors. However, the way the adaptive immune response - T cells in particular - functions in Salmonella cancer therapies remains largely unknown, and previous studies in mice suggest that T cells do not play any role in the efficacy of Salmonella-mediated cancer therapy. Here, we aimed at understanding how T cells function in Salmonella cancer therapy using mouse models of CRC.

Results

Tumours treated with an attenuated Salmonella typhimurium strain (STm Δ aroA) exhibited impaired activation of both CD4+ and CD8+ T cells. T cells activated in the presence of STm-infected tumours showed reduced proliferation and production of cytokines in ex vivo tumors and co-culture models with tumor organoids. T cell receptor signalling was initially intact, but full activation could not be sustained due to an inability to induce metabolic reprogramming. This was underpinned by the failure to sustain c-Myc protein levels, and c-Myc inhibition was driven by Salmonella-induced asparagine depletion. Removing Salmonella asparaginase activity restored asparagine levels and T cell activity. This, however, came at the expense of suppressing tumour cell c-Myc activities and effects on the tumour stem cell compartment. Despite this, use of STm Δ aroA also lacking asparaginase reduced tumour burden in a model of colitis-associated CRC.

Impact

This work has improved our understanding of T cell functionality during STm cancer therapy and is further applicable to other bacterial strains. Our findings will contribute to the rational engineering of STm-cancer therapies and co-therapy indications.

6/ Thank you for providing a synopsis, however please note that the stand-first should be 300 characters maximum, including spaces (see examples <https://www.embopress.org/journal/17574684>). Please adjust accordingly.

Please also suggest a visual abstract to illustrate your article as a PNG file 550 px wide x 300-600 px high. A cropped portion of this image will serve as thumbnail for the table of content on our webpage.

7/ As part of the EMBO Publications transparent editorial process initiative (see our Editorial at <http://embomolmed.embopress.org/content/2/9/329>), EMBO Molecular Medicine will publish online a Review Process File (RPF) to accompany accepted manuscripts.

This file will be published in conjunction with your paper and will include the anonymous referee reports, your point-by-point response and all pertinent correspondence relating to the manuscript.

Please note that reviews from the previous journal will NOT be included, therefore please remove this part from your point-by-point rebuttal letter.

Let us know whether you agree with the publication of the RPF and as here.

I look forward to receiving your revised manuscript.

Yours sincerely,

Lise Roth

The authors addressed the remaining editorial issues.

11th Oct 2024

Dear Dr. Maslowski,

Thank you for submitting your revised files. I am pleased to inform you that your manuscript is accepted for publication and is now being sent to our publisher to be included in the next available issue of EMBO Molecular Medicine.

Please note that I introduced minor modifications in your synopsis text, let me know if you agree with the following:

"Attenuated Salmonella show promise as cancer therapeutics, yet T cells play no role in efficacy, limiting combined checkpoint blockade therapies or induction of immune memory. Improving T cell responses requires a better understanding of the causes of T cell dysfunction during Salmonella BCT.

- Attenuated Salmonella enterica Typhimurium (STm) cancer therapy drives T cell dysfunction
- STm depletes asparagine via an asparaginase (AnsB) in the tumour microenvironment (TME), suppressing tumour growth
- Asparagine depletion in the TME triggers T cell metabolic arrest by destabilising c-Myc
- An STm Δ ansB mutant restores T cell function but at the cost of enhanced tumour metabolism
- The STm Δ ansB mutant still effectively reduces tumour growth in a mouse model"

We have also resized your visual abstract and cropped a small portion (115x70 pixels) to serve as a thumbnail for the table of content on our webpage (both attached). Let us know as soon as possible if changes are needed, as further modification during proofing are usually not allowed.

With kind regards,

Lise Roth
